# Small extracellular vesicles in follicular fluids for predicting reproductive outcomes in assisted reproductive technology

Ayako Muraoka[1], Akira Yokoi [1,2,3] ✉, Kosuke Yoshida[1,2], Masami Kitagawa[4], Eri Asano-Inami[1], Mayuko Murakami[1], Bayasula[4], Natsuki Miyake [1], Natsuki Nakanishi[1], Tomoko Nakamura[1], Satoko Osuka[1], Akira Iwase[5] & Hiroaki Kajiyama[1]

## Abstract

**Background** Assisted reproductive technology accounts for an increasing proportion of infertility treatments, and assessments to predict clinical pregnancy outcomes are desired. Extracellular vesicles exist in follicular fluid, and small non coding RNAs in extracellular vesicles underline the possibility of reflecting pregnancy potential.

**Methods** Follicular fluid samples are collected from 20 ovarian follicles of 15 infertile patients undergoing assisted reproductive technology. Extracellular vesicles are isolated by serial centrifugation and small RNA sequencing is performed to investigate the profiles of microRNAs and P-element-induced wimpy testis-interacting RNAs.

**Results** Small extracellular vesicles with a size range of approximately 100 nm are successfully isolated, and the small non coding RNA profiles of pregnant samples ($n = 8$) are different from those of non-pregnant samples ($n = 12$). Fourteen dysregulated small non coding RNAs are selected to identify the independent candidates [mean read count >100, area under the curve >0.8]. Among them, we find that a specific combination of small non coding RNAs (miR-16-2-3p, miR-378a-3p, and miR-483-5p) can predict the pregnant samples more precisely using a receiver operating characteristics curves analysis (area under the curve: 0.96). Furthermore, even in the same patients, the three microRNAs are differentially expressed between pregnant and non-pregnant samples.

**Conclusions** Our results demonstrate that small non coding RNAs derived from small extracellular vesicles in follicular fluid can be potential non-invasive biomarkers for predicting pregnancy, leading to their probable application in assisted reproductive technology. Further large-scale studies are required to validate the clinical usefulness of these small non coding RNAs.

## Plain language summary

Assisted reproductive technologies (ART) are medical procedures used to address infertility. Follicular fluid (FF)is a liquid that surrounds the immature egg cells (oocytes) and provides hormones and nutrients necessary for their maturation and eventual development into embryos. We analyzed genetic components within the FF as a potential predictor of reproductive outcomes following ART. Here, we show that a specific combination of genetic elements produced by the FF in successful pregnancies did not occur in unsuccessful pregnancies. Our findings may help to provide a non-invasive approach to determining reproductive outcomes during ART procedures.

The demand for assisted reproductive technology (ART) is increasing with the increasing age of first childbearing age[1]. The International Committee for Monitoring Assisted Reproductive Technologies (ICMART) annual world report series in 2014 mentioned that an estimated two million cycles of in vitro fertilization (IVF) and embryo transfer (ET) were conducted worldwide[2]. ART has benefitted infertile patients. However, the pregnancy rate per treatment is approximately 20%; thus, ART is repeated multiple times until pregnancy[2]. The response of ovarian function to hormonal stimulation or the implantation rate of embryo transfer is important for improving the outcomes of pregnancy. Currently, the ovarian reserve is estimated based on the serum concentration of anti-Müllerian hormone (AMH)[3]. However, the AMH level is insufficient for predicting future

pregnancies with ART. Therefore, novel predictive biomarkers are highly anticipated to improve the pregnancy rates with ART.

The quality of embryos, which is critical for successful pregnancy, is primarily assessed by morphological scoring, including the number of blastomeres and the rate of fragmentation[4]. However, this morphological scoring system relies on the experience of embryologists and there are no clinically available objective indicators. Follicular fluid (FF) provides a complex and suitable environment for oocyte maturation and contains several molecules secreted from the oocyte, granulosa, cumulus, and theca cells[5–7]. Previous studies have shown that specific microRNAs (miRNAs) in FF are associated with embryo quality in human beings[8]. However, they have not discussed the relationship between the miRNAs and pregnancy outcomes.

miRNAs are small, single-stranded RNAs comprising a length of approximately 22 nucleotides[9]. miRNAs exist in various body fluids, such as the peripheral blood, urine, and follicular fluid[10]. In these fluids, miRNAs show remarkable stability, especially when enclosed within membrane-based nanovesicles, such as extracellular vesicles (EVs), including exosomes[11]. EVs are known to contain various molecules such as messenger RNAs (mRNAs), proteins, and miRNAs, which are actively released into the extracellular space, and can play important roles in cell-to-cell communication by transferring their molecules to target cells[11,12]. Several scientists have examined the functional role of intracellular communication via EVs. The miRNAs in FF-EVs have recently been identified as a mechanism of intercellular communication and as a source of biomarkers[13]. In addition to miRNAs, P-element-induced wimpy testis (PIWI)-interacting RNAs (piRNAs) are also major types of small non coding RNAs (ncRNAs) that are 21–35 nucleotides long and work together with PIWI proteins[14,15]. In mammalian oocytes and early embryos, piRNAs play important roles in protecting the germline of animals by silencing transposons and regulating gene expression[16,17]. Recently, Yang et al. described a highly sensitive single-cell small RNA sequencing method to profile small ncRNAs in human oocytes and early embryos[18]. Previous studies have shown that deficiencies in the piRNA pathways lead to female sterility in zebrafish[19]. Based on these reports, it can be hypothesized that piRNAs also play an essential role in the human germline, including FF.

Here, we investigate the small ncRNAs in FF-EVs as a predictive biomarker for the pregnancy potential related to ART. We successfully isolate FF-EVs and find that a specific combination of small ncRNAs (miR-16-2-3p, miR-378a-3p, and miR-483-5p) can predict the pregnant samples more precisely. Furthermore, functional analysis of these candidate miRNAs reveals the association of follicular development and embryo quality. Our findings provide valuable insight into oocyte development and maturation during folliculogenesis.

## Methods
### Study population
The FF samples were collected from women undergoing ART at Nagoya University Hospital. Informed consent was obtained from each patient prior to ovarian stimulation. Consent to publish clinical information potentially identifying individuals (e.g., age, gender, clinical history, etc.) was obtained. This study was approved by the Ethics Committee of Nagoya University School of Medicine (approval number:2022-0010). FF from single follicles containing mature oocytes were obtained and analyzed. Multiple follicles were punctured from individual patient and FF samples were obtained from 299 retrieved oocyte, and 33 samples resulted in pregnancy and delivery (Fig. 1a). Of these 33 samples, 8 samples were included in the pregnant group after excluding samples in which the cause of infertility was a male factor and tubal factor ($n = 14$), as well as cases in which two embryos were transferred simultaneously ($n = 4$), cases in which there was insufficient sample volume ($n = 5$) and cases in which there was insufficient embryo data ($n = 2$). Twelve embryos from the non-pregnant samples that seemed to match the maternal age and grade of the transferred embryos were selected as the non-pregnant group (Supplementary Fig. S1). In this study, "pregnant" means childbirth without chemical and clinical pregnancies.

The term "non-pregnant" means a negative pregnancy reaction (HCG negative by urine or blood tests) after embryo transfer. Patients with endometriosis or polycystic ovarian syndrome (PCOS) as an apparent cause of infertility were excluded.

### Ovarian stimulation and the IVF/intracytoplasmic sperm injection (ICSI) procedure
Ovarian stimulation involves the administration of urinary follicle-stimulating hormone (FSH) (uFSH Aska, ASKA Pharmaceutical Co., Ltd., Tokyo, Japan) or recombinant FSH (Gonalef, Merck BioPharma, Tokyo, Japan) at 150–300 IU per day for the first two days, after which the doses were adjusted individually based on the follicular response under gonadotropin-releasing hormone antagonist or agonist protocols. The prevention of premature luteinizing hormone (LH) surge was achieved by daily administration of ganirelix acetate (GANIREST, Organon & Co., Jersey City, NJ, USA) or nafarelin acetate hydrate (Nasanyl Nasal Spray 0.2%, Pfizer, NY, USA). When the follicles reached 18 mm or more in mean diameter, 10,000 IU of human chorionic gonadotropin (HCG for injection, Fuji Pharmaceutical Co. Inc., Toyama, Japan) was administered, and transvaginal oocyte retrieval was performed 35.5 h later. All sperm preparation and IVF/ICSI protocols were performed similarly at our hospital[20]. ICSI or conventional methods were performed depending on the sperm concentration and motility.

### Embryo transfer (ET)
In the ET protocol, all the patients were treated using the same HRT protocol. A transdermal estradiol patch (Estrana® tape 0.72 mg; Hisamitsu Pharmaceutical Co. Inc. Tokyo, Japan) was used to stimulate endometrial growth, and progestin vaginal capsules (UTROGESTAN vaginal capsules 200 mg; Fuji Pharmaceutical Co. Inc. Tokyo, Japan; or LUTINUS vaginal tablet 100 mg; Ferring Pharmaceuticals Co., Ltd.) was used for luteal-phase support. All the Gardner and Veeck classification of the transferred embryo were 3CB or better, G3 or better, respectively (Supplementary Fig. S1).

### Sample collection
FF were obtained from follicles measuring 18–25 mm in diameter from each patient. We collected each oocyte and FF, individually. Washing was performed after each puncture to prevent contamination between follicular samples. The FF sample volume of each individual follicle ranges from 2 to 10 mL. All FF samples were individually placed into 15-mL conical tubes and centrifuged at speed ($430 \times g$) for 10 min, aliquoted into 2-mL tubes of clear supernatant, and stored at −80 °C until the analysis.

### Isolation of EVs
The EV isolation method used in this study adhered to the standard principles of the International Society for Extracellular Vesicles[21,22]. Following the removal of cellular debris through centrifugation at $430 \times g$, approximately 1 mL of each FF sample was further centrifuged at $10,000 \times g$ for 40 min at 4 °C to eliminate medium and large-sized EVs using a Kubota Model 7000 centrifuge (KUBOTA Co., Tokyo, Japan). The supernatant was filtered using a 0.22 μm filter (Millex-GV 33 mm, Millipore) and then ultracentrifuged at $110,000 \times g$ for 70 min at 4 °C using an MLS50 rotor (Beckman Coulter Inc., USA). The pellet was washed with PBS, ultracentrifuged under the same conditions, and resuspended in PBS to extract the small EVs (sEVs). The protein concentrations of EVs and cell lysates were quantified using the Qubit protein assay kit (Thermo Fisher Scientific) with a Qubit 4.0 Fluorometer (Invitrogen Co., MA, USA), according to the manufacturer's protocol. The size distribution and particle concentration in the EV preparations were analyzed using a NanoSight NS300 (Malvern Panalytical Ltd., UK) nanoparticle tracking analyzer. The samples were diluted in PBS and injected at a speed of 100 a.u. into the measuring chamber, and the EVs flow was recorded in triplicate (30 s each) at room temperature. The equipment settings for data acquisition were kept constant between measurements, with the camera level set to 13.

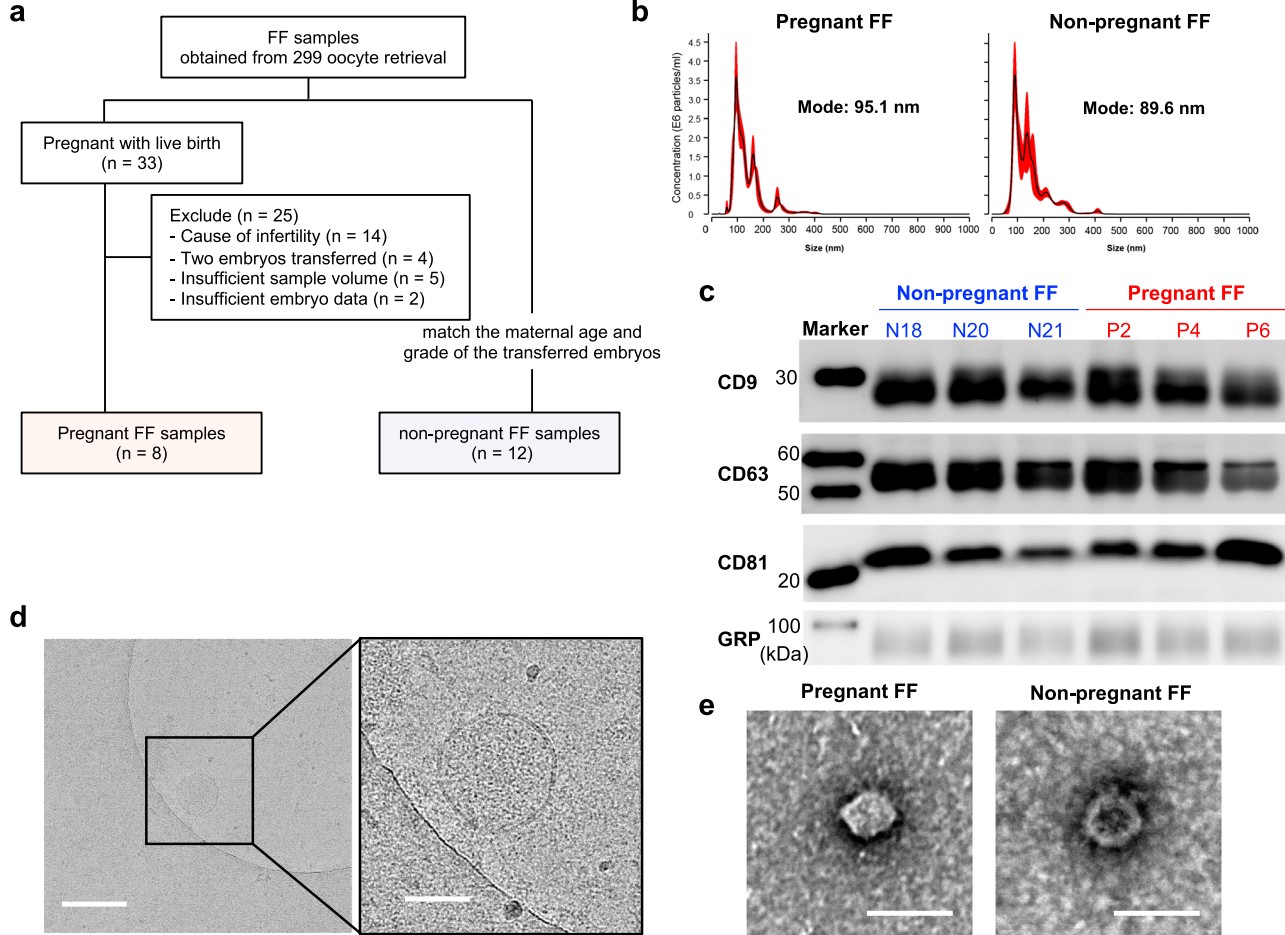

Fig. 1 | **Characterization and expression analysis of FF-sEVs. a** Flowchart of the process for the FF sample selection. **b** Size distribution of FF-sEVs determined by a nanoparticle tracking analysis. **c** The protein expression of EV markers (CD9, CD63, and CD81) and cytoplasmic marker of GRP in FF-sEVs. P means samples in pregnant groups and N means samples in non-pregnant groups. Morphology of the FF-sEVs detected by Cryo-transmission electron microscopic imaging (**d**) and transmission electron microscopy (**e**). The scale bars represent 100 nm.

## Cryo-transmission electron microscopic imaging (cryo-TEM)

The morphology of FF-sEVs was visualized using a cryo-transmission electron microscope (Terabase Inc., Okazaki, Japan) that could generate high-contrast images of the nanostructures of biological materials without requiring staining procedures that could damage the samples. The natural structure of the sample distributed in the solution was observed by preparing the sample using a rapid vitreous ice-embedding method. The assays were performed according to previously published methods[23]. For cryo-electron microscopic experiments, 2.5 uL of each sample solution was applied to standard Copper Quantifoil grids R1.2/1.3 (Quantifoil Micro Tools GmbH, Germany) and vitrified by rapid plunging into liquefied ethane using a Vitrobot Mark IV (Thermo Fisher Scientific) at 95% humidity and 4 °C. The frozen grid was then mounted on a cryo-transfer specimen holder (Model626, Gatan, CA, USA) at liquid nitrogen temperature and loaded into a JEM-2200FS microscope (JEOL, Tokyo, Japan), equipped with a field emission gun and an omega-type energy filter, operated at 200 kV. A slit width of 15 eV was used to obtain a zero-energy-loss electron beam. Images were recorded using a DE-20 direct electron detection device (Direct Electron LP, CA, USA) at 30,000× magnification, corresponding to a pixel size of 1.96 Å on the specimen.

## Western blotting analysis of EVs and cell lysates

The samples of FF-sEVs prepared with adjusted amounts of protein were loaded onto polyacrylamide gels for electrophoretic separation of proteins at 20 mA. After blocking with Skim Milk (Snow Brand Megmilk Co., Japan) or Blocking One (Nacalai Tesque Inc., Japan) for 1 h at room temperature, the membranes were incubated overnight at 4 °C with the following primary

antibodies: mouse monoclonal anti-CD9 (CBL162; Merck; dilution 1:100), rabbit monoclonal anti-CD63 (EXOAB-CD63A-1; System Biosciences, LLC, CA, USA; dilution 1:1000), mouse monoclonal anti-CD81 (sc-166029; Santa Cruz Biotechnology, TX, USA; dilution 1:100), and mouse monoclonal anti-GRP (sc-393402; Santa Cruz Biotechnology; dilution 1:100). The membranes were subsequently washed three times for 5 min each using Tris-buffered saline with 0.1% Tween® 20 (TBST) and incubated for 1–3 h at room temperature with secondary HRP-conjugated mouse anti-rabbit IgG (NA934-1ML; Cytiva Lifesciences, USA; dilution 1:5000) or anti-mouse IgG (NA931-1ML; Cytiva; dilution 1:2000) antibodies. The protein ladder is MagicMark™ XP Western Protein Standard (Thermo Fisher Scientific). The membranes were imaged using ImageQuant LAS 4010 (GE Healthcare, IL, USA). The uncropped blots are shown in Supplementary Fig. S2.

## Small RNA sequencing

The RNAs were extracted using the miRNeasy Plus Mini Kit (QIAGEN, Hilden, Germany). The total RNA concentration of each sample was measured using the Agilent RNA6000 pico Kit and Agilent 2100 Bioanalyzer (Agilent Technologies, Santa Clara, CA, USA). We conducted measurements to verify that there were no differences in RNA concentration between the pregnancy and non-pregnancy groups (range: 73–998 pg/uL, and 92–1194 pg/μL, respectively). RIN values of FF-EV RNAs were around 1.0–3.0. Small RNA libraries were prepared using the NEBNext Multiplex Small RNA Library Prep Set for Illumina (New England Biolabs, Ipswich, MA, USA) from 6 μL RNA. Index codes were added to attribute sequences to each sample. Next, the PCR products were purified using the QIAquick PCR

Purification Kit (Qiagen) and 6% TBE gel (120 V, 60 min). Furthermore, DNA fragments corresponding to 140–160 bp (the length of small RNA plus the 3′ and 5′ adaptors) were recovered, and the concentration of complementary DNAs (cDNAs) was measured using a Qubit dsDNA HS Assay Kit and a Qubit2.0 Fluorometer (Life Technologies, Carlsbad, CA, USA). Finally, single-end reads were obtained using Illumina MiSeq or NextSeq (Illumina, San Diego, CA, USA). The raw data files of small RNA sequencing were analyzed using CLC Genomics Workbench version 9.5.3 (Qiagen). After the adaptor trimming, the data were mapped to miRBase 22 and piRBase v3.0, allowing up to two mismatches. As a result, 1157 miRNAs and 984,769 piRNAs were annotated (Supplementary Fig. S3). Differentially expressed ncRNAs were analyzed using the DEseq2 package (ver. 1.36.0) in RStudio (RStudio, Boston, MA, USA) with R software (ver. 4.2.1). Using a volcano plot, differentially expressed ncRNAs were determined using the criteria of $P < 0.01$ and an absolute log2 fold change >0.8. Heatmap and principal component analysis (PCA) were performed for the 430 differentially expressed ncRNAs. miRNA and piRNA profiles were normalized separately using reads per million mapped reads, and the heatmap.2 function in the gplots package (ver. 3.1.3) and plot3d functions of the rgl package (ver. 0.110.2) were used. Moreover, receiver-operating characteristics (ROC) curves were generated using the qROC package (ver. 1.18.0), and the propensity score was calculated using a logistic regression model. Furthermore, the corrplot package (version 0.92) was used to visualize the correlation of expression.

### Quantitative polymerase chain reaction (qPCR)
Briefly, cDNAs were synthesized using the TaqMan™ Advanced miRNA cDNA Synthesis Kit (Thermo Fisher Scientific) according to the manufacturer's instructions. TaqMan® Fast Advanced Master Mix (Thermo Fisher Scientific) and TaqMan™ Advanced miRNA Assay (Assay ID 477931_mir for hsa-miR-16-2-3p, Assay ID 478349_mir for hsa-miR-378a-3p, Assay ID 478432_mir for hsa-miR-483-5p, Assay ID 483023_mir for hsa-miR-1246, and Assay ID 477895_mir for hsa-miR-1290; Thermo Fisher Scientific) were used. Then, qPCR was performed using an Mx3000P (Agilent Technologies). The PCR conditions consisted of an initial denaturation step at 95 °C for 10 min, followed by 40 amplification cycles of 95 °C for 15 s and 60 °C for 1 min. The amplified product was monitored by measuring the increase in FAM fluorescence intensity. Each experiment was performed in triplicate and repeated at least three times to ensure reproducibility.

### Statistics and reproducibility
We used miRSystem (http://mirsystem.cgm.ntu.edu.tw/) to perform functional annotation to predict target mRNAs. Gene Ontology (GO) analysis was performed by using DAVID (https://david.ncifcrf.gov/tools.jsp); notably, $P < 0.05$ were set as the cutoff.

Microsoft Excel and R were used to generate graphs and to perform statistical analyses. $P$ values are indicated as n.s., $P > 0.05$, $*P < 0.05$, $**P < 0.01$. Data were analyzed by either a two-tailed Student's $t$ test, Welch's $t$ test, or Fisher's exact test. The statistical methods used in this study are described in each Figure legend and Table note.

### Reporting summary
Further information on research design is available in the Nature Portfolio Reporting Summary linked to this article.

## Results
### The characterization of FF-EVs
FF samples with pregnancy ($n = 8$) and FF samples without pregnancy ($n = 12$) were included in this study (Fig. 1a). Each FF sample was collected individually from each follicle at the time of oocyte retrieval and stored until the experiments. We divided the stocked FF samples into two groups: the pregnant group, comprising samples from individuals who achieved pregnancy after subsequent ET, and the non-pregnant group, consisting of samples from individuals who did not achieve pregnancy after ET. There were no significant differences in age, body mass index, cause of infertility, duration of infertility, ART method, or ET between the two groups (Table 1).

**Table 1 | Patient characteristics in FF cohort**

| Variable | Pregnant FF ($n = 8$) | Non-pregnant FF ($n = 12$) | $P$ value |
|---|---|---|---|
| Age, years (±SD) | 35 ± 3.9 | 35 ± 3.2 | 0.94[a] |
| BMI (± SD) | 22 ± 2.4 | 21 ± 2.6 | 0.73[a] |
| Diagnosis | | | |
| Primary infertility | 6 | 7 | 0.64[b] |
| Secondary infertility | 2 | 5 | |
| Duration of infertility, years | | | |
| <5 | 7 | 12 | 0.33[b] |
| 6 to 10 | 1 | 0 | |
| >10 | 0 | 0 | |
| ART method | | | |
| IVF | 1 | 3 | 0.62[b] |
| ICSI | 7 | 9 | |
| Embryo transfer | | | |
| Fresh | 1 | 3 | 0.62[b] |
| Frozen | 7 | 9 | |

*FF* follicular fluid, *BMI* body mass index, *ART* assisted reproductive technology, *IVF* in vitro fertilization, *ICSI* intracytoplasmic sperm injection, *SD* standard deviation.
[a]Student $t$ test.
[b]Fisher's exact test.

The FF-EVs were isolated by serial centrifugation methods, and a nanoparticle tracking analysis showed that the size of the particles was less than 200 nm in diameter (Fig. 1b). Therefore, the FF-EVs in this study mainly consisted of sEVs, including exosomes. Moreover, western blot analyses showed that FF-sEVs were positive for major EV markers, including CD9, CD63, and CD81, and negative for GRP (Fig. 1c). Additionally, cryo-electron microscopic analyses and TEM confirmed the presence of sEVs (Fig. 1d, e). Based on these data, we successfully isolated sEVs from FF.

### Identifying predictive biomarkers by small RNA sequencing for FF-sEVs
Subsequently, we performed small RNA sequencing, and the sequence data were mapped to miRNA and piRNA references. The detailed flowchart of analysis method is described in Supplementary Fig. S3, and a total of 1157 miRNAs and 984,769 piRNAs were annotated (Fig. 2a). The volcano plot showed that 3 miRNAs were significantly upregulated in FF-sEVs in samples from pregnant women, whereas 10 miRNAs were downregulated. Similarly, 195 piRNAs were significantly upregulated in FF-sEVs in pregnant groups, whereas 222 piRNAs were downregulated (Fig. 2b, Supplementary Table S1 and Supplementary Data 1). Then, we performed the heatmap and PCA using the 430 differentially expressed ncRNAs. The ncRNA profiles of the pregnant and non-pregnant groups were clearly clustered into different clusters (Fig. 2c, d). Subsequently, to narrow down the small ncRNA candidates, the AUC of each small ncRNA was calculated. The baseline expression levels of each small ncRNA were also taken into consideration. As a result, 2 upregulated miRNAs, five downregulated miRNAs, zero upregulated piRNAs, and seven downregulated piRNAs were selected (mean read count >100, AUC > 0.8; Fig. 2e and Supplementary Fig. S4). Finally, to identify the independent candidates, the correlation coefficients of the 14 small ncRNAs were calculated (Fig. 2f). From each cluster, we selected miR-16-2-3p, miR-378a-3p, and miR-483-5p because they had showed a relatively higher AUC and baseline expression (Fig. 2e, f). Hence, the predictive accuracy of the combination of these three small ncRNAs was evaluated. The propensity score was calculated using a logistic regression model. The ROC curve showed that the combination of the three small ncRNAs was more accurate (AUC 0.96; Fig. 2g). Additionally, the propensity score was significantly different between the pregnant and non-pregnant groups ($P < 0.01$; Fig. 2h).

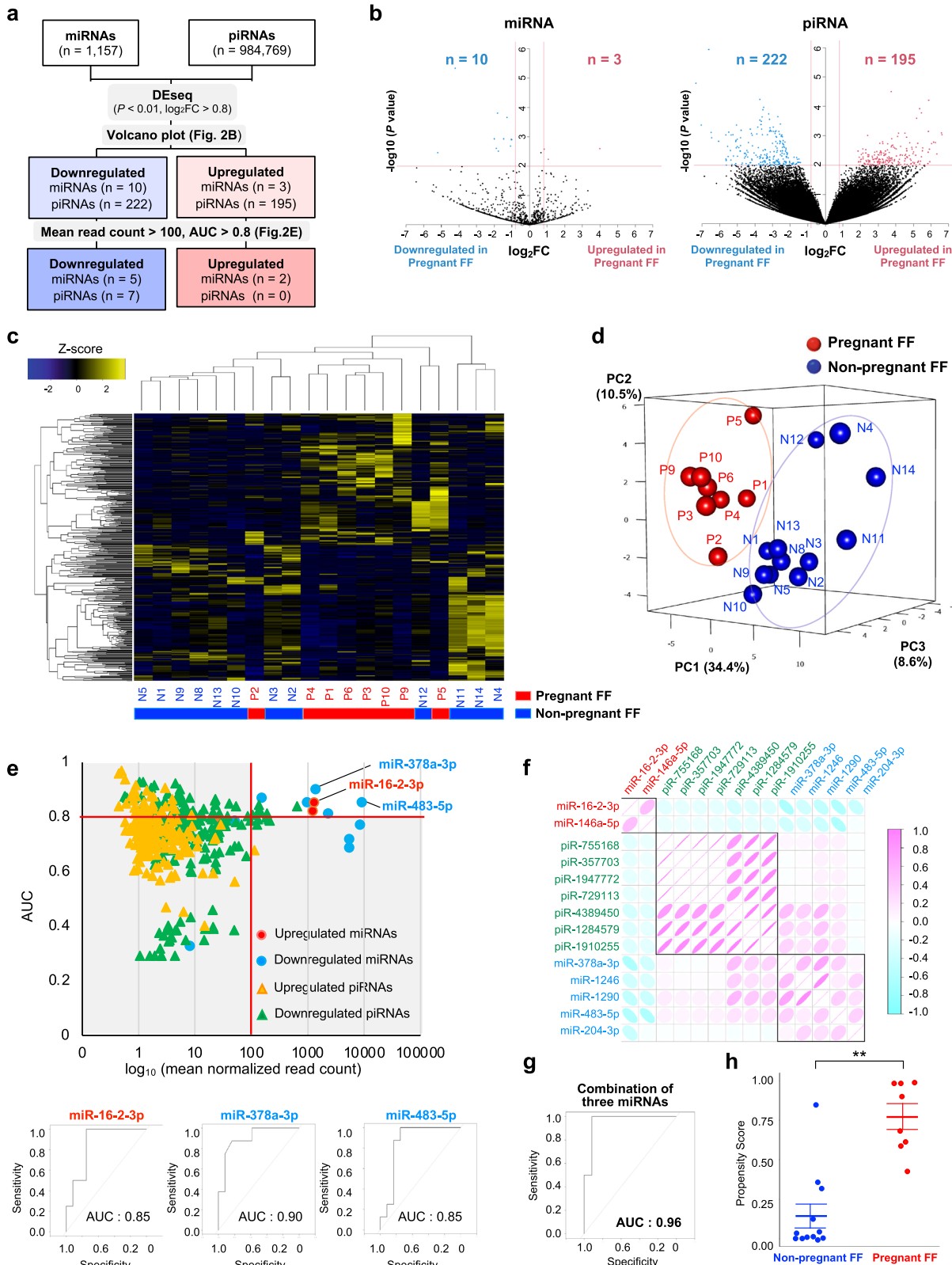

In addition, the expression of these miRNAs (miR-16-2-3p, miR-378a-3p, miR-483-5p, miR-1246, and miR-1290) in FF-sEVs was confirmed using qPCR. The levels of miR-16-2-3p was significantly higher in the pregnant group than in the non-pregnant group. However, for the other four miRNAs, although there were no significant differences, we observed trends in their expression levels (Supplementary Fig. S5).

## Comparison of pregnant and non-pregnant cases in FF-sEVs in each patient

In our cohort, three patients had both pregnant and non-pregnant oocytes, and we compared the small RNA profiles of FF-sEVs. The patient characteristics are shown in Table 2. The heatmap and PCA revealed that the pregnant FF-sEVs had similar small ncRNA profiles, regardless of

**Fig. 2 | Selecting small ncRNAs in FF-sEVs as biomarkers. a** Flowchart of the process for the small ncRNA selection. **b** Volcano plot of the small ncRNAs. The log2FC, *P* values, and IfcSE were calculated using the DEseq2 package (ver. 1.36.0). **c** The heatmap analysis for 430 differentially expressed ncRNAs in FF-sEVs. The normalized data were converted to base 10 logarithms and z-score. P means samples in pregnant groups and N means samples in non-pregnant groups. **d** The principal component analysis of the 430 ncRNAs. Blue and red circles indicate the non-pregnant FF and pregnant FF, respectively. PC means principal component. **e** Dot plots showing the performance of the 430 differentially expressed ncRNAs, and ROC curves for miR-16-2-3p, miR-378a-3p, and miR-483-5p. Red and blue circles

indicate the upregulated and downregulated miRNAs, respectively. Red and blue circles indicate the upregulated and downregulated miRNAs, respectively. Yellow and green triangles indicate the upregulated and downregulated piRNAs, respectively. AUC means area under the curve. **f** Correlation diagram of the 14 small ncRNAs. **g** ROC curve of the combination of the three small ncRNAs (miR-16-2-3p, miR-378a-3p, and miR-483-5p). **h** Dot plot of the propensity score calculated using the three small ncRNAs (miR-16-2-3p, miR-378a-3p, and miR-483-5p). \*\*$P < 0.01$. The data were analyzed by the Welch's *t* test. The pregnant group: $n = 8$, non-pregnant group: $n = 12$, respectively, biologically independent samples.

## Table 2 | Patient characteristics in FF sub-cohort

| No. | FF-EVs | Outcome | Age at OPU, years | BMI | Diagnosis | Duration of infertility, years | ART method |
|---|---|---|---|---|---|---|---|
| Patient 1 | P2 | pregnant | 30–40 | 23.9 | Secondary infertility | 1 | IVF |
| | N10 | non-pregnant | | | | | |
| | N11 | non-pregnant | | | | | |
| Patient 2 | P4 | pregnant | 30–40 | 25.1 | Primary infertility | 0.5 | ICSI |
| | P5 | pregnant | | | | | |
| | N12 | non-pregnant | | | | | |
| Patient 3 | P10 | pregnant | 30–40 | 20.3 | Primary infertility | 3 | ICSI |
| | N13 | non-pregnant | | | | | |

*FF* follicular fluid, *OPU* oocyte pick up, *BMI* body mass index, *ART* assisted reproductive technology, *IVF* in vitro fertilization, *ICSI* intracytoplasmic sperm injection.

individual differences (Fig. 3a, b). The heatmap analysis revealed a tendency for samples from the same patients to cluster within the same group, whereas the PCA indicated that pregnancy status could influence the location of samples. For instance, in patient 1, the small ncRNA profile of pregnant FF-sEVs-P2 was different from that of non-pregnant FF-sEVs-N10 and -N11 (Fig. 3a). Therefore, small ncRNA profiles of FF-sEVs may reflect the oocyte quality. Next, we investigated the expression of these three small ncRNA candidates (Fig. 3c). The lower levels of miR-378a-3p and miR483-5p were possible to be considered as biomarkers for pregnancy in these three patients. Moreover, higher levels of miR-16-2-3p can be associated with pregnancy in patients 2 and 3.

### Functional analysis for miRNAs

Finally, functional analysis was performed to determine whether the extracted miRNAs are actually associated with follicular development. First, the miRSystem algorithm was used to assess the potential functions of the seven differentially expressed miRNAs (miR-16-2-3p, miR-146a-5p, miR-204-3p, miR-378a-3p, miR-483-5p, miR-1246, and miR-1290) shown in Fig. 2e. Among these miRNAs, we focused on three selected miRNAs (miR-16-2-3p, miR-378a-3p, and miR-483-5p), which we found to be associated with 189 mRNAs according to miRSystem analysis. Second, we selected 640 mRNAs associated with embryo quality from a previous study[24] and integrated 189 mRNAs related to the selected miRNAs, resulting in narrowed-down set of seven genes (Fig. 3d). These seven mRNAs included genes that play critical roles in embryonic development, such as *SOX* and *RHO*. GO terms were significantly dysregulated by the seven miRNAs, with notable impact on the BMP signaling pathway, which is known as important pathway for oocyte development (Fig. 3d). These results imply that these miRNAs present in FF-sEVs may be associated with both embryo quality and follicular development.

### Discussion

The quality of IVF embryos is directly related to the pregnant outcomes, and FF provides a suitable microenvironment for growing mature oocytes. FF-EVs are required for intercellular communication between the oocytes and surrounding cells and may be a biological marker for predicting pregnant outcomes. In this study, we identified differentially expressed small ncRNAs in FF-sEVs of pregnant samples compared to non-pregnant samples. A

specific combination of these small ncRNAs in FF-sEVs could predict pregnancy in a more precise manner. The FF microenvironment is known to reflect the oocyte status and the relationships between FF-EVs and oocyte maturation in human and bovine[25–28], molecular mechanisms of PCOS[29] and age[30–32] have been reported in previous studies. A more precise study revealed the peptide profile of biomarkers associated with fertilization outcome[33]. However, an association between pregnancy outcomes and FF-EV-derived small ncRNAs has not been reported. This study evaluated the usefulness of small ncRNAs as non-invasive biomarkers for pregnancy outcomes using comprehensive small RNA sequencing.

miRNAs are the major regulators of gene expression and are involved in the regulation of diverse biological processes, including development, proliferation, and differentiation. In the context of reproduction, dynamic changes in miRNA expression have been reported in FF, granulosa cells, cumulus cells, and oocytes, indicating their critical functions during follicular development and oocytes maturation[34–36]. miRNAs present in EVs also play important roles in reproductive processes, such as fertilization, implantation, and embryogenesis[37]. EVs facilitate intracellular communication by transporting molecules, including non-cording RNAs, and contribute to exchange of materials between cells. In this study, we identified several miRNAs related to pregnancy outcomes, and particularly, the combination of three miRNAs (miR-16-2-3p, miR-378a-3p, and miR-483-5p) had the potential for more precise prediction of pregnancy outcomes. Previous studies have reported downregulation of miR-483-5p in conditioned medium from the pregnant group[38] and in granulosa cells from groups with good-quality embryos and live births[39], which suggested that miR-483-5p plays an important role in embryo development. miR-378a-3p has been identified as a hatching-associated miRNA in a previous report[40], although our results are contradictory. However, this discrepancy indicates that miR-378a-3p may have a directional role in the oocyte maturation, and further validation may reveal its potential critical function. There are no reproductive reports on miR-16-2-3p, underscoring the need for further studies.

Follicular development usually involves multiple pathological changes, suggesting that a single biomarker may not be sufficient to reflect the entire process. Thus, a combination of biomarkers is usually required to enhance the predictive efficiency of pregnancy outcomes. In this study, we demonstrated that the diagnostic capability of three small ncRNAs (miR-16-2-3p, miR-378a-3p, and miR-483-5p) from the multivariate logistic regression

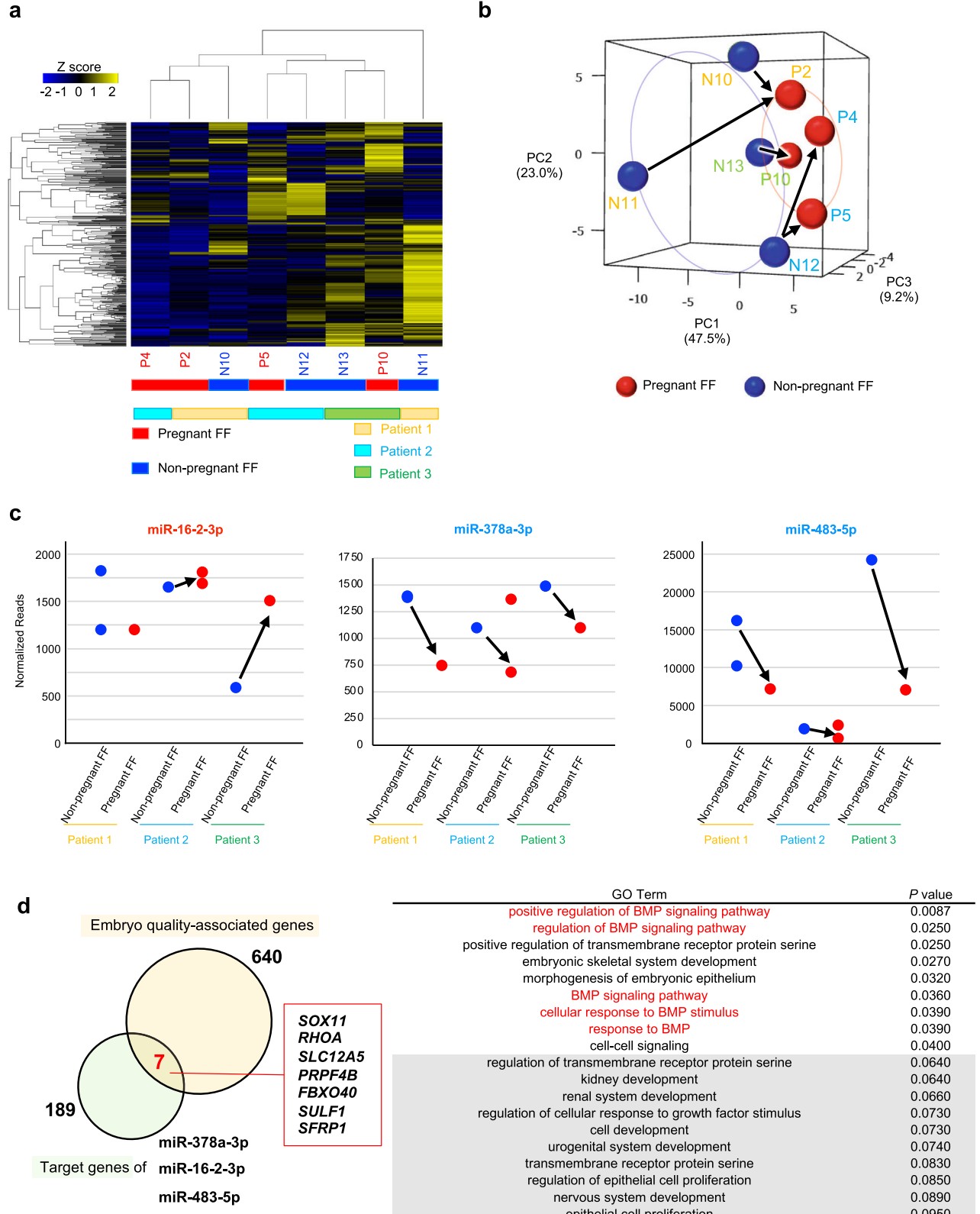

**Fig. 3 | The expression of small ncRNAs in the eight FF-sEVs from three patients.**
**a** The clustering and heatmap analysis of the small ncRNA profiling in FF-sEVs.
**b** The PCA of the small ncRNA profiling shown in (**a**). The arrows indicate the association between the FF-sEVs with non-pregnant and pregnant in each patient. Blue and red circles indicate the non-pregnant FF and pregnant FF, respectively. PC means principal component. **c** The normalized read counts of miR-16-2-3p, miR-378a-3p, and miR-483-5p. The arrows indicate the association between the FF-sEVs with non-pregnant and pregnant in each patient. Blue and red circles indicate the non-pregnant FF and pregnant FF, respectively. **d** Functional annotation of the selected three miRNAs (miR-16-2-3p, miR-378a-3p, and miR-483-5p) and embryo quality-associated genes. Yellow and green circles indicate the embryo quality-associated genes and the target genes of miR-378a-3p, miR-16-2-3p, and miR-483-5p, respectively. GO terms were listed with *P* values.

model showed better predictive efficacy than any single small ncRNA. Moreover, we compared the small ncRNA profiles of FF-sEVs in the pregnancy and non-pregnancy samples from the same individuals. These results suggest that candidate small ncRNAs in FF-sEVs may be biological markers that can predict pregnancy across individual differences. Therefore, the small ncRNAs identified in our study show great potential as non-invasive biomarkers for pregnancy. However, "non-invasive" here refers to the use of already collected FF as a by-product for biomarkers, without further invasive procedures, despite the invasiveness of oocyte retrieval.

In addition, further functional analysis is required for miRNAs and piRNAs, since they are considered to have equivalent predictive roles as biomarkers and vital functions in FF. In this study, we showed that differentially expressed miRNAs might contribute to embryo quality and follicular development. miRNAs can simultaneously regulate the expression levels of many target genes involved in folliculogenesis by binding to partially complementary sequences and resulting in subsequent interference with mRNA stability. On the contrary, the functions of piRNAs have not yet been elucidated. Notably, piRNAs work together with PIWI proteins and act by maintaining transposons in a repressed methylated state in germline cells[41]. Maintaining genome integrity in germline cells ensures an accurate transmission of genetic information throughout generations. In zebrafish, disruption of the piRNA pathway causes both female and male sterility[42]. Further studies are warranted to reveal the functions of piRNAs in humans.

This study had several limitations. First, we could only select a relatively small number of pregnancy samples because of low pregnancy outcomes. Further large-scale studies are required to obtain samples from multiple institutions. Second, although some of the miRNAs found in this study provided a convincing explanation for the differentially expressed miRNAs and pregnancy, others exhibited dysregulation in the opposite direction in different studies. Differential endpoints, container of miRNAs, technical differences in sample handling, RNA extraction, normalization, and statistical analysis methods are likely to account for at least some of the differences between studies.

In conclusion, this report shows the use of small ncRNA biomarkers in FF-sEVs for predicting pregnancy using comprehensive, highly sensitive small RNA sequencing. Small ncRNAs in FF-sEVs can predict pregnancy outcomes, especially when using a combination of small ncRNAs. Our findings could contribute to potential noninvasive pregnancy predictors that are clinically applicable to ART.

## Data availability

The small RNA sequence data have been deposited in the Gene Expression Omnibus (GEO) under accession number GSE221271. The source data for the figures can be found in the source data files (Supplementary Data 2, including the source data of Fig. 2e to h, Supplementary Figs. S4, and S5b), and all other data are available from the corresponding author on reasonable request.

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

## Acknowledgements

We express our gratitude to the members of the Department of Obstetrics and Gynecology at the Nagoya University Graduate School of Medicine. We received technical support from the Division of Medical Research Engineering of Nagoya University Graduate School of Medicine. We would like to thank Editage (www.editage.com) for English language editing. This work was financially supported by a Grant-in-Aid for Scientific Research, the Japan Society for the Promotion of Science (JSPS KAKENHI Grant Numbers 21H03075 and 22K16832), the Fusion Oriented Research for Disruptive Science and Technology (FOREST; JPMJFR204J) from the Japan Science and Technology Agency, and Tokai Pathways to Global Excellence (T-GEx), part of MEXT Strategic Professional Development Program for Young Researchers. Moreover, the Daiichi Sankyo Foundation of Life Science, Mochida Memorial Foundation for Medical and Pharmaceutical Research, and Uehara Memorial Foundation also supported this study.

## Author contributions

Conceptualization: A.M., A.Y. Methodology: A.Y., K.Y., M.K., E.A.-I., M.M. Investigation: M.K., M.M., B.B., K.Y., N.M., N.N., T.N. Funding acquisition: A.Y. Project administration: A.Y., S.O. Supervision: A.Y., S.O., A.I., H.K. Writing – original draft: A.M., A.Y. Writing – review and editing: A.M., A.Y., K.Y., and S.O. All the authors discussed the results and commented on the manuscript.

## Competing interests

The authors declare no competing interests.

## Additional information

[1]Department of Obstetrics and Gynecology, Nagoya University Graduate School of Medicine, 65 Tsurumai-cho, Showa-ku, Nagoya 466-8550, Japan. [2]Nagoya University Institute for Advanced Research, Furo-cho, Chikusa-ku, Nagoya 464-8603, Japan. [3]Japan Science and Technology Agency (JST), FOREST, 4-1-8 Honcho, Kawaguchi, Saitama 332-0012, Japan. [4]Bell Research Center for Reproductive Health and Cancer, Nagoya University Graduate School of Medicine, 65 Tsurumai-cho, Showa-ku, Nagoya 466-8550, Japan. [5]Department of Obstetrics and Gynecology, Gunma University Graduate School of Medicine, 3-39-22 Showa-machi, Maebashi 371-8511, Japan. ✉e-mail: ayokoi@med.nagoya-u.ac.jp

