## [Peer Review File · Communications Medicine]

Reviewers' comments:

Reviewer #1 (Remarks to the Author):

In the presented study, the authors propose using follicular fluid-derived small EVs (and thereby their content) as biomarkers to predict the chances of success for patients undergoing fertility treatment in assisted reproductive technology. The current method of measuring the Anti-Müllerian hormone, which indicates the ovarian reserve, is an insufficient and possibly incorrect parameter to predict the outcome of the fertility treatment. Hence the authors claim that there is a need for better markers to help clinicians and patients realistically assess the chances of success in the event of repeated ART treatments, multiple times.

The paper does need some further improvements to make the author's points stronger. Below I list the points which I think need to be considered before publication:

1) miRNA's can also be cell-free and there are studies focusing on miRNAs in FF. Why have the authors chosen to work with sEV miRNA ? would it not be easier to just look for cell-free miRNA?

2) There are additionally studies that have studied the peptides and proteins in FF and suggested using them as biomarkers to predict better outcomes.

It would be useful to compare this work to work done in the past focusing on FF components used as biomarkers in ART and highlight the advantage of the proposed method.

e.g. <https://proteomesci.biomedcentral.com/articles/10.1186/s12953-016-0106-9>

<https://www.nature.com/articles/srep25486>

- Santonocito, M. et al. Molecular characterization of exosomes and their microRNA cargo in human follicular fluid: bioinformatic analysis reveals that exosomal microRNAs control pathways involved in follicular maturation. *Fertil Steril* 102, 1751–1761 e1751 10.1016/j.fertnstert.2014.08.005 (2014)

3) Line 91: Grammatical error. I think it should be EVs are known to be the carriers (and not cargo) of several...

4)Line 221: In order to make it easier for the reader to follow, it would help if the authors explained a little more in detail what they mean by FF with or without pregnancy. Was the FF sample collected prior to the ART procedure in both cases?

5) Fig 1d: The western blot is presented in an unconventional manner. Even in the supplementary figure, there is no labeling of which well has what sample. There was no figure legend as well

6) Line 248: Why was miR-483-5p chosen and not miR-378a-3p? Downregulation of miR-483-5p has been shown by others to have a negative correlation with pregnancy outcome.. (so this observation is not entirely novel..) also miR-378 shows better AUC values..? Please explain and make it more clear in the article text.

7) L258: the authors state that fig3a shows the similarity between small RNA profiles among pregnant FF-sEVs (which is not so obvious) so kindly explain.. the same figure is also referred to, to indicate the differences in the small RNA profiles for the same patient (which is observable). It

would have helped to just have the figure comparing the profiles of each patient side by side. Also, what the no.2 / 19/20 mean has not been adequately explained. table 2 please correct the spelling of patient.

8) Line 263-264 : please rephrase for clarity in statement.

9) Discussion lacks a speculation on the individual miRNAs or piRNAs and their known function and how that correlates to the observations made. The discussion also fails to highlight the advantage of using EVs from FF as a biomarker source. Please expand the discussion to include these topics.

Reviewer #2 (Remarks to the Author):

The manuscript “follicular fluid-derived small extracellular vesicles for predicting successful fertility treatment in assisted reproductive technology” investigated the presence of small RNAs in extracellular vesicles from ovarian follicular fluid from patients undergoing IVF treatment. The experimental design is very well described. The material and methods are well described and the analysis are well performed. However, the manuscript needs English editing and lacks novelty since others have performed the same or similar studies. I would suggest to add new analysis or some sort of functional experiments besides just descriptive analysis. Data can be further explored in the search for new transcripts, biological pathways and biological functions.

Line 33: “FF samples were collected from 24 oocytes”? Or were collected from 24 ovarian follicles?

Line 54: Please revise English format.

Lines 301-303: “In this study, we showed that differentially expressed miRNAs might contribute to germline cell replication using an in silico analysis.” I did not understand this sentence.

Reviewer #3 (Remarks to the Author):

Manuscript reference COMMSMED-22-0508

The Manuscript entitled “Follicular fluid-derived small extracellular vesicles for predicting successful fertility treatment in assisted reproductive technology” submitted by Muraoka et al., to Communications Medicine (reference COMMSMED-22-0508) investigates the potential of specific small non-coding RNA cargo contained in extracellular vesicles (EVs) derived from follicular fluid from patients undergoing fertility treatments to predict the pregnancy outcomes. The manuscript covers a hot topic in the field of reproduction and there is an increasing interest in the identification of RNA EV cargo from different reproductive fluids as potential biomarkers of fertility and pregnancy, which could be great assets in ART success. However, the manuscript has some major flaws that are addressed below.

1) Starting material and isolation of EVs

The methodology needs to be clarified and the following details added to the manuscript:

- FF volume: Which range of volumes of FF from each follicle were collected? It is not clear if the EV analysis was performed from FF from 1 unique follicle and in this case, which was the volume taken for all samples? It was possible to take the same volume from all patients (1 ml)?
- Assessment of RNA concentration: What was the RNA concentration obtained from each sample? Please, add the range of RNA concentration for pregnant and non-pregnant samples? Were differences in the total RNA concentration between pregnant and non-pregnant samples?
- Assessment of RNA quality: Were the EV RNA profiles analyzed by Bioanalyzer to assess the similarity of RNA profiles among pregnant and non-pregnant groups and in general across samples? (It seems only RNA concentration by Qubit was analyzed).
- Which was the total RNA amount was used as starting material for the library preparation?
- Regarding the method for isolation of EV, in which protocol were the authors based?
- Why samples were first centrifuge into 15-mL conical tubes and centrifuged at speed (15,000 rpm) for 10 min?
- The authors mentioned that their EV isolation protocol followed the standard of International Society of EVs and mentioned two references: Witwer et al., 2013 and They et al., 2018. These publications are guidelines for EVs studies but do not describe a detail protocol for successful FF EV isolation. Why did the authors not used an already established protocol for FF EVs? Why the isolation protocol starts with 10,000g. Usually, serial centrifugations, (300 g, followed by 2000 and 12,000) are used to remove cellular and cellular debris.

2) Characterization of EVs:

- It is not clear if the Cryo-TEM analysis was performed on a of pool sample or individual replicates were analyzed. Where pregnant and non-pregnant analyzed in separated? Furthermore, close-up images should be accompanied by wide-field views to allow assessment of heterogeneity (according to MISEV) in both type of samples.
- Nanoparticle tracking analysis: Again here, was this analysis performed on pool samples or individual replicates? Were pregnant and non-pregnant samples analyzed in separate to determine differences in EV size distribution or EV concentration between pregnant and non-pregnant samples?
NTA results should be consistent: In the abstract, it is stated that small EVs (sEVs) had a size range of approximately 100 nm; then in the results: EV have less than 200nm (line 225) and then NTA Figure 1B shows Mode 88 nm. Please, explain how many samples were analyzed by NTA, show the range of EV sizes and re-write the results.
- Exosomal markers: Three tetraspanins (CD9, CD63 and CD81 have been used as EV markers, but at least 1 or 2 exosomal cytoplasmic markers (TSG101, ALIX) should have been included (as MISEV guidelines advice).
- Marker of contamination: Ultracentrifugation was used to isolate EV and although has been considering the gold standard method it can yield a less pure EV population that other methods (SEC, sucrose gradient). Thus, a marker of contamination could provide some clues about the purity of the EV pellets. For example, GP96, endoplasmic reticulum marker, could have been used in the Western Blot to prove its presence in cell lysates and absence in EV samples, indicating absence of other cellular membrane contamination in the EV preparations.
Regarding the cell lysates, it seems the protein concentration was measured but they were not used for WB or were they?
- Western Blot (WB) results: Although the authors provided the complete blots of the WB results (uncropped blots in Supplementary Figures), the blots seem to be cut (or maybe the gels and the

blots are shown, but pictures are confusing). There is a lot of more missing information in these blots: 1) the molecular weight in the ladder bands; 2) In the 3 blots, there are 2 sample wells: What are the differences among wells? (EVs and cell lysates). If these data are only the blots, as it is stated, the biggest concern is that CD63 and CD9 antibodies do not seem very specific (multiple bands).

- Are WB performed on pool samples (pregnant and non-pregnant together)?
- How much protein was loaded in the gels?

3) Analysis of RNA-seq data

- The analysis of the data does not seem very appropriate, given the results presented. Why the author did not perform Edge R or DESeq2 analyses as most studies with this kind of data do? Why did not do correction for multiple Test, which will provide a most strict criterion to identify differentially abundant (DA) RNAs among samples?

Instead, why a Welch's t-test was performed? The cut off criterion of P-value <0.025 and $|\log_2FC| >0.7$ is not very strict.

Suggestion: To re-do the analysis with these potential options and present the data in the heatmaps and PCA with only the DA miRNAs or piRNAs.

-All miRNAs and piRNAs detectable or identified in the study should be provided in the manuscript. Besides, every list showing the filtering of the miRNA and piRNA data together with the final lists of RNA showing higher or lower abundance after statistical analysis (FDR, FC, Pvalues) should be provided. The manuscript only shows 2 lists of selected miRNAs and piRNAs (with a very unclear filtering criteria).

-What was the size of the libraries for all samples? Was there any variability?

- Data filtering is unclear: It is stated that data was filtered to exclude small RNAs with low read (miRNAs, <100 reads in all samples; piRNAs, <300 reads in all samples). However, in the filtered list provided in the paper, miRNAs and piRNAs with lower than 100 reads are listed. Next filtering performed to narrow the list of selected RNAs is even more unclear and an appropriate shorten list is not shown (based on Pvalue, reads and AUC).

4) Results of RNA-seq data

In Figure 1 (E_H), the heatmaps and PCA do not show two RNA profiles for pregnant and non-pregnant samples as the authors stated (Not for miRNAs and not for piRNAs). The authors seem to describe what they would like to see.

Given the PCA and heatmap plots that the authors showed, they could have discussed a potential grouping of samples in pregnant and non-pregnant with the exception of a few samples, which could be the ones shown in the Fig1 E (heatmap).

Suggestion: After performing another RNA-seq data analysis based on a more strict criterion and presenting only the DA RNAs in a PCA and heatmap, if there are still samples that they do not group together, they could be discussed as: 1) potential outliers or 2) maybe as samples that they could provide a favorable environment for an oocyte, which could have good quality but because other reasons (male, chance) these oocytes finally do not end in a successful pregnancy.

Please, in PCA and heatmaps write the samples labels, which allow to identify which samples do not group together.

5) Selected miRNAs and piRNAs:

Did the authors consider the function of these selected miRNAs and piRNA and the potential target genes that could act in the oocyte and affects its quality and have an impact on the potential embryo? It is not discussed at all in the manuscript. For example, hsa-miR-483-5p, was previously found in human FF EVs (Martinez et al., 2018 10.1038/s41598-018-35379-3).

6) Results of cohort study (3 patients) and selection of biomarkers:

The heatmap in Fig.3 (A) does not show two different RNA profiles between pregnant and non-

pregnant women (Even the clustering three does not group samples in 2 groups). Besides, the results for the 3 selected RNAs showed a lot of variability among patients and do not reflect the results stated by the authors (Fig 3. (C)). It is difficult to see how they could be pointed as biomarkers, with these results and without discussing their potential functions and target genes.

Again here, Why the authors do not illustrate only the DA miRNAs, which could show the biggest differences among samples.

Suggestion: If the goal is to suggest RNAs as biomarkers, another technique together with another cohort of samples, could have been used to validate the results of these selected miRNAs (for example by Taq-man assays for specific miRNAs)

7) Data mining

This part is very superficial with only 5 pathways shown and with a very brief discussion. Suggestion: Other available online tools could be used to find potential target genes for the differentially abundant miRNAs and associated functional enrichment analysis. Combination of results from different functional enrichment analysis can confirm results or bring different potential perspectives.

8) Comparison with other studies.

Did the authors compared the miRNAs and piRNAs identified in this study with others?

Which was the overlap? It is stated that variability among studies has been discussed in the manuscript (Lines 312 to 317), but it is not mentioned which comparison has been done and which small noncoding RNAs were found in common to other studies.

9) Datasets uploaded to repository data

Datasets should be uploaded to repository data (for example: <https://www.ncbi.nlm.nih.gov/geo/>).

10) Methodology:

Methodology should be described in a way that experiments or techniques can be repeat it by others. For example, description of preparation of samples for CryoTEM analysis should be improved or references added.

11) Figures legend: Legend should be written with all necessary information to understand the heatmap and PCA. For example: What is z-score?

Minor concerns:

-Some sentence needs to be re-written: "EVs are known to be the cargo of several mRNAs, proteins, and miRNAs". Please change to "are known to contain..."

-Please change small RNA for small non coding RNAs (small ncRNAs) across the manuscript.

Reviewer #1 (Remarks to the Author):

In the presented study, the authors propose using follicular fluid-derived small EVs (and thereby their content) as biomarkers to predict the chances of success for patients undergoing fertility treatment in assisted reproductive technology. The current method of measuring the Anti-Müllerian hormone, which indicates the ovarian reserve, is an insufficient and possibly incorrect parameter to predict the outcome of the fertility treatment. Hence the authors claim that there is a need for better markers to help clinicians and patients realistically assess the chances of success in the event of repeated ART treatments, multiple times. The paper does need some further improvements to make the author's points stronger. Below I list the points which I think need to be considered before publication:

Thank you for your valuable feedback. In response to your comments, we have carefully revised the article.

1) miRNA's can also be cell-free and there are studies focusing on miRNAs in FF. Why have the authors chosen to work with sEV miRNA? would it not be easier to just look for cell-free miRNA?

Answer: Thank you for your comment. Cell-free miRNA generally comprises all miRNAs present in body fluids and can also include miRNA released passively from surrounding cells, including dying cells. In contrast, EVs are actively released into the extracellular space and play important roles in cell-to-cell communication by transferring proteins, RNA, and miRNA molecules to target cells. In this study, we focused on small RNAs present in FF-EVs to identify functionally relevant nucleic acids produced by granulosa cells. We have made the necessary additions to the Introduction section to clarify these points of view (Page 4, Lines 89–92).

2) There are additionally studies that have studied the peptides and proteins in FF and suggested using them as biomarkers to predict better outcomes.

It would be useful to compare this work to work done in the past focusing on FF components used as biomarkers in ART and highlight the advantage of the proposed method.

e.g. <https://proteomesci.biomedcentral.com/articles/10.1186/s12953-016-0106-9>

<https://www.nature.com/articles/srep25486>

- Santonocito, M. et al. Molecular characterization of exosomes and their microRNA cargo in human follicular fluid: bioinformatic analysis reveals that exosomal microRNAs control pathways involved in follicular maturation. Fertil Steril 102, 1751–1761 e1751 10.1016/j.fertnstert.2014.08.005 (2014)

Answer: Thank you for the valuable references. As suggested, we have incorporated them into the revised manuscript and accordingly revised the Discussion section (Page 14, Lines 338–343).

3) Line 91: Grammatical error. I think it should be EVs are known to be the carriers (and not cargo) of several...

Answer: We have corrected the error as follows “EVs are known to contain” (Page 4, Lines 89–90).

4)Line 221: In order to make it easier for the reader to follow, it would help if the authors explained a little more in detail what they mean by FF with or without pregnancy. Was the FF sample collected prior to the ART procedure in both cases?

Answer: Thank you for your important comment. As pointed out, we have carefully revised the sentences to clarify FF with or without pregnancy (Page 11, Lines 259–263).

5) Fig 1d: The western blot is presented in an unconventional manner. Even in the supplementary figure, there is no labeling of which well has what sample. There was no figure legend as well

Answer: Thank you for your comment. We have re-performed the analysis. Detailed descriptions have also been added in the revised version.

6) Line 248: Why was miR-483-5p chosen and not miR-378a-3p? Downregulation of miR-483-5p has been shown by others to have a negative correlation with pregnancy outcome. (So, this observation is not entirely novel.) also miR-378 shows better AUC values...? Please explain and make it more clear in the article text.

Answer: We apologize for any confusion. We had primarily focused on miR-483-5p because of its higher normalized read count in the small RNA sequencing than that of miR-378a-3p in the original version. However, we taken into consideration the feedback from another reviewer and have conducted a thorough RNA sequencing re-analysis. As a result of the re-analysis, miR-378a-3p was also detected as a candidate miRNA.

7) L258: the authors state that fig3a shows the similarity between small RNA profiles among pregnant FF-sEVs (which is not so obvious) so kindly explain. the same figure is also referred to, to indicate the differences in the small RNA profiles for the same patient (which is observable). It would have helped to just have the figure comparing the profiles of each patient side by side. Also, what the no.2 / 19/20 mean has not been adequately explained. table 2 please correct the spelling of patient.

Answer: We apologize for the confusion and typographical errors in Table 2. We have carefully revised the Results and corrected them.

8) Line 263-264: please rephrase for clarity in statement.

Answer: Thank you for this comment. We have made corrections based on your comment.

9) Discussion lacks a speculation on the individual miRNAs or piRNAs and their known function and how that correlates to the observations made. The discussion also fails to highlight the advantage of using EVs from FF as a biomarker source. Please expand the discussion to include these topics.

Answer: Thank you for your helpful comment. We further analyzed the functions of the miRNAs revealed in our study and verified their relevance to embryo quality and follicular development. We have carefully revised the Results and Discussion sections as follows.

- Page 13, Lines 317–326 (Result section)

First, the miRSystem algorithm was used to assess the potential functions of the seven differentially expressed miRNAs (miR-16-2-3p, miR-204-3p, miR-378a-3p, miR-424-5p, miR-483-5p, miR-1246, and miR-1290) shown in Fig 2E. Among these miRNAs, we focused on three selected miRNAs (miR-16-2-3p, miR-378a-3p, and miR-483-5p), which were found to be associated with 189 mRNAs according to miRSystem analysis. Second, we selected 640 mRNAs associated with embryo quality from a previous study (PMID: 36857405) and integrated 189 mRNAs related to the selected miRNAs, resulting in narrowed-down set of seven genes (Fig. 3D). These seven mRNAs included genes that play critical roles in embryonic development, such as SOX and RHO. Gene ontology (GO) terms were significantly dysregulated by the seven mRNAs, with notable impact on the BMP signaling pathway (Fig. 3D).

- Page 14, Lines 344–361 (Discussion section)

miRNAs are the major regulators of gene expression and are involved in the regulation of diverse biological processes, including development, proliferation, and differentiation. In the context of reproduction, dynamic changes in miRNAs expression have been reported in FF, granulosa cells, cumulus cells, and oocytes, indicating their critical functions during follicular development and oocytes maturation (PMID: 19296935, PMID: 30630485, PMID: 23904466). miRNAs present in EVs also play important roles in the reproductive processes, such as fertilization, implantation, and embryogenesis (PMID: 26663221). EVs facilitate intracellular communication by transporting molecules, including non-coding RNAs, and contribute to exchange of materials between cells. In this study, we identified several miRNAs related to pregnancy outcomes, and particularly, the combination of three miRNAs (miR-16-2-3p, miR-378a-3p, and miR-483-5p) had

the potential for more precise prediction of pregnancy outcomes. Previous studies have reported downregulation of miR-483-5p in conditioned medium from the pregnant group (PMID: 34988157) and in granulosa cells from groups with good-quality embryos and live births (PMID: 36411450), which suggested that miR-483-5p plays an important role in embryo development. miR-378a-3p has been identified as a hatching-associated miRNA in a previous report (PMID: 35298333), although our results are contradictory. However, this discrepancy indicates that miR-378a-3p may have a directional role in oocyte maturation, and further validation may reveal its potential critical function. There are no reproductive reports on miR-16-2-3p, underscoring the need for further studies.

Reviewer #2 (Remarks to the Author):

The manuscript “follicular fluid-derived small extracellular vesicles for predicting successful fertility treatment in assisted reproductive technology” investigated the presence of small RNAs in extracellular vesicles from ovarian follicular fluid from patients undergoing IVF treatment. The experimental design is very well described. The material and methods are well described and the analysis are well performed. However, the manuscript needs English editing and lacks novelty since other have performed the same or similar studies. I would suggest to add new analysis or some sort of functional experiments besides just descriptive analysis. Data can be further explored in the search for new transcripts, biological pathways, and biological functions.

Thank you for your positive feedback on our data. Based on your suggestions, we have thoroughly revised the article. We focused on extracellular vesicles present in follicular fluid (FF-EVs). Our study is novel because the endpoint is pregnancy, whereas other studies have solely assessed embryo quality. Additionally, we have carefully reviewed and proofread the English language throughout the manuscript, adding English language proof certification. Furthermore, we have included the functional analysis of miRNAs revealed in our study and verified their relevance to embryo quality and follicular development.

Line 33: “FF samples were collected from 24 oocytes”? Or were collected from 24 ovarian follicles?

Answer: We sincerely apologize for the confusion. We have carefully revised the sentence as “FF samples were collected from 24 ovarian follicles of 19 infertile patients undergoing ART.”

Line 54: Please revise English format.

Answer: Thank you for your comment. We have carefully revised the sentence as follows: Subsequently, we investigated the expression of small RNAs, including ncRNAs such as miRNAs and piRNAs, using comprehensive small RNA sequencing.

Lines 301-303: “In this study, we showed that differentially expressed miRNAs might contribute to germline cell replication using an in silico analysis.” I did not understand this sentence.

Answer: We apologize for the confusion. We have carefully revised the sentence (Page 15, Lines 373–377).

Reviewer #3 (Remarks to the Author):

Manuscript reference COMMSMED-22-0508

The Manuscript entitled “Follicular fluid-derived small extracellular vesicles for predicting successful fertility treatment in assisted reproductive technology” submitted by Muraoka et al., to Communications Medicine (reference COMMSMED-22-0508) investigates the potential of specific small non-coding RNA cargo contained in extracellular vesicles (EVs) derived from follicular fluid from patients undergoing fertility treatments to predict the pregnancy outcomes. The manuscript covers a hot topic in the field of reproduction and there is an increasing interest in the identification of RNA EV cargo from different reproductive fluids as potential biomarkers of fertility and pregnancy, which could be great assets in ART success. However, the manuscript has some major flaws that are addressed below.

Thank you for your valuable feedback. To address the important points you raised, we have substantially revised our manuscript as much as possible and deeply appreciate your help in improving this manuscript.

1) Starting material and isolation of EVs

The methodology needs to be clarified and the following details added to the manuscript:

- FF volume: Which range of volumes of FF from each follicle were collected? It is not clear if the EV analysis was performed from FF from 1 unique follicle and in this case, which was the volume taken for all samples? It was possible to take the same volume from all patients (1 ml)?

Answer: We sincerely apologize for the insufficient writing. We have carefully revised the sentence and added “The FF sample volume of each individual follicle ranges from 2 to 10 mL” in the Materials and Methods section (Page 7, Lines 146–147).

- Assessment of RNA concentration: What was the RNA concentration obtained from each sample? Please, add the range of RNA concentration for pregnant and non-pregnant samples? Were differences in the total RNA concentration between pregnant and non- pregnant samples?

Answer: We apologize for the inadequate data. We have carefully revised the Methods section and added the range of RNA concentrations (Page 9, Lines 204–207). We have also added data showing that there was no difference in total RNA concentration between the two groups (Supplementary Fig. S4B).

-Assessment of RNA quality: Were the EV RNA profiles analyzed by Bioanalyzer to assess the similarity of RNA profiles among pregnant and non-pregnant groups and in general across

samples? (It seems only RNA concentration by Qubit was analyzed).

Answer: We apologize for the inadequate data. RNA concentrations were measured using a Bioanalyzer and all data were added to Supplementary Fig. S4B. While RNA concentrations were low, these results are reasonable because they detected small non-coding RNAs in the FF-sEVs. In addition, only a limited amount of FF sample could be collected, however, RNA sequencing has the advantage that RNA concentration can be amplified for analysis. Even RNA in FF at low concentrations can be a target for study by sequencing using amplification methods.

-Which was the total RNA amount was used as starting material for the library preparation?

Answer: We apologize for the inadequate data. As shown in the additional data on bioanalyzers, RNA concentrations were approximately 0.1–1 ng/μL, and we used 6 μL of RNA for library preparations. We understand that this concentration is low, but we confirmed that sequencing worked well for analysis based on replication assessments (data not shown).

- Regarding the method for isolation of EV, in which protocol were the authors based?

Answer: We apologize for the insufficient writing. Our group routinely performs serial centrifugation methods following ISEV indications, as reported in several previous studies (PMID: 26563733, PMID: 24009892, PMID: 28262727, PMID: 25828099).

- Why samples were first centrifuge into 15-mL conical tubes and centrifuged at speed (15,000 rpm) for 10 min?

Answer: We noticed that we had made a mistake in the description. We routinely perform centrifugations at 1,500 rpm for 10 min, and we have revised this point in the updated manuscript (Page 7, Line 148).

- The authors mentioned that their EV isolation protocol followed the standard of International Society of EVs and mentioned two references: Witwer et al., 2013 and They et al., 2018. These publications are guidelines for EVs studies but do not describe a detail protocol for successful FF EV isolation. Why did the authors not used an already established protocol for FF EVs? Why the isolation protocol starts with 10,000g. Usually, serial centrifugations, (300 g, followed by 2000 and 12,000) are used to remove cellular and cellular debris.

Answer: We understand that the references are not detailed protocols, but they include many valuable tips that must be considered. First, we apologize for the mistake in the description of FF storage. The samples were centrifuged at 1,500 rpm for 10 min to remove cellular debris. Our group has been performing serial centrifugation methods recommended by MISEV2018 and reported in several previous studies (PMID: 26563733, PMID: 24009892, PMID: 28262727,

PMID: 25828099). As known by most EV researchers, we must be careful about EV diversity, and we performed an initial centrifugation step at 10,000 ×g for 40 min to remove medium and large EVs (PMID: 26858453). After 10K centrifugation, we filtered the samples using a 0.22-µm filter. Based on these steps, our samples consisted mainly of small EVs, such as exosomes. To the best of our knowledge, there is no gold standard for FF-EV isolation, but our modified method precisely follows the concept of serial centrifugations for isolating small EVs.

2) Characterization of EVs:

- It is not clear if the Cryo-TEM analysis was performed on a of pool sample or individual replicates were analyzed. Where pregnant and non-pregnant analyzed in separated? Furthermore, close-up images should be accompanied by wide-field views to allow assessment of heterogeneity (according to MISEV) in both type of samples.

Answer: For cryo-TEM, the analysis was performed using 3 mL of pool sample of FF from three patients. According to your suggestions, wide-field views have been added (Fig. 1D, left panel). Unfortunately, it was difficult to show the heterogeneity of the samples due to low concentrations. However, NTA data showed their diversity in sizes, and the peak size of EVs was approximately 100 nm. To assess the morphological difference of FF-EV from pregnant and non-pregnant cases, we additionally performed TEM using samples from those cases. As shown in Fig. 1E, they look similar.

- Nanoparticle tracking analysis: Again here, was this analysis performed on pool samples or individual replicates? Were pregnant and non-pregnant samples analyzed in separate to determine differences in EV size distribution or EV concentration between pregnant and non-pregnant samples?

NTA results should be consistent: In the abstract, it is stated that small EVs (sEVs) had a size range of approximately 100 nm; then in the results: EV have less than 200nm (line 225) and then NTA Figure 1B shows Mode 88 nm. Please, explain how many samples were analyzed by NTA, show the range of EV sizes and re-write the results.

Answer: NTA was performed using individual replicates. At the beginning of this project, we assessed three samples from each group, and the data were found to be consistent. In addition, this size is consistent with the additional TEM data. In the revised version, representative NTA results are shown in Fig. 1B and TEM data are shown in Fig. 1E.

-Exosomal markers: Three tetraspanins (CD9, CD63 and CD81 have been used as EV markers, but at least 1 or 2 exosomal cytoplasmatic markers (TSG101, ALIX) should have been included (as MISEV guidelines advice).

-Marker of contamination: Ultracentrifugation was used to isolate EV and although has been considering the gold standard method it can yield a less pure EV population than other methods (SEC, sucrose gradient). Thus, a marker of contamination could provide some clues about the purity of the EV pellets. For example, GP96, endoplasmic reticulum marker, could have been used in the Western Blot to prove its presence in cell lysates and absence in EV samples, indicating absence of other cellular membrane contamination in the EV preparations.

Regarding the cell lysates, it seems the protein concentration was measured but they were not used for WB or were they?

-Western Blot (WB) results: Although the authors provided the complete blots of the WB results (uncropped blots in Supplementary Figures), the blots seem to be cut (or maybe the gels and the blots are shown, but pictures are confusing). There is a lot of more missing information in these blots: 1) the molecular weight in the ladder bands; 2) In the 3 blots, there are 2 sample wells: What are the differences among wells? (EVs and cell lysates). If these data are only the blots, as it is stated, the biggest concern is that CD63 and CD9 antibodies do not seem very specific (multiple bands).

-Are WB performed on pool samples (pregnant and non-pregnant together)?

- How much protein was loaded in the gels?

Answer: We completely re-performed WB using samples from pregnant and non-pregnant patients and not pooled samples. We applied 2 µg of samples to each lane. The cytoplasmic marker of GRP was also assessed. We used antibodies that were widely used in previous studies. As expected, GRP was not expressed in the FF-EV samples. We understand that there were some non-specific bands, and targeted bands were observed. The data are shown in Fig. 1C and Supplementary Fig. S1.

3) Analysis of RNA-seq data

- The analysis of the data does not seem very appropriate, given the results presented. Why the author did not perform Edge R or DESeq2 analyses as most studies with this kind of data do? Why did not do correction for multiple Test, which will provide a most strict criterion to identify differentially abundant (DA) RNAs among samples?

Instead, why a Welch's t-test was performed? The cut off criterion of P-value <0.025 and $|\log_2FC| > 0.7$ is not very strict.

Suggestion: To re-do the analysis with these potential options and present the data in the heatmaps and PCA with only the DA miRNAs or piRNAs.

Answer: We have completely re-conducted the analysis according to your suggestions. We investigated differentially expressed ncRNAs using the DESeq2 package. In the re-analysis, we determined the cutoff criterion of P-value as <0.01 and $|\log_2FC|$ as >0.85. As a result, two

upregulated miRNAs, 15 downregulated miRNAs, 123 upregulated piRNAs, and 289 downregulated piRNAs were selected. We agreed that this criterion is not so strict. However, a lower P-value does not necessarily indicate a better biomarker. Moreover, strict criteria may overlook attractive biomarkers. We attempted to use an adjusted P-value for the criteria, but only a few candidates were extracted. Therefore, we believe that this criterion was appropriate for this study.

-All miRNAs and piRNAs detectable or identified in the study should be provided in the manuscript. Besides, every list showing the filtering of the miRNA and piRNA data together with the final lists of RNA showing higher or lower abundance after statistical analysis (FDR, FC, Pvalues) should be provided. The manuscript only shows 2 lists of selected miRNAs and piRNAs (with a very unclear filtering criteria).

Answer: Thank you for your comment. As shown in Supplementary Fig. S2, we annotated 1,212 miRNAs and 1,076,849 piRNAs. We can provide both miRNA and piRNA profiles. However, the piRNA profile was less informative because the read counts of most piRNAs were very low. Instead, we have provided a list of selected miRNAs and piRNAs as Supplementary Tables, including the average read count of the pregnancy and non-pregnancy groups, area under the ROC curve, and P-value calculated using DEseq2.

-What was the size of the libraries for all samples? Was there any variability?

Answer: Thank you for your comment. First, the median sequence read was 10,513,675 reads (range, 3,648,701–15,935,439), and there was no significant difference between the pregnant and non-pregnant groups. The median annotated ncRNAs was 2,520,754 reads (range, 710,564–7,305,291), and the median annotation rate was 25.90% (17.14%–45.84%). Moreover, there were no significant differences between the pregnant and non-pregnant groups regarding annotated ncRNAs and annotation rate. Overall, we do not believe that there is a significant difference with respect to the libraries.

- Data filtering is unclear: It is stated that data was filtered to exclude small RNAs with low read (miRNAs, <100 reads in all samples; piRNAs, <300 reads in all samples). However, in the filtered list provided in the paper, miRNAs and piRNAs with lower than 100 reads are listed. Next filtering performed to narrow the list of selected RNAs is even more unclear and an appropriate shorten list is not shown (based on Pvalue, reads and AUC).

Answer: We have completely redone the analysis according to your suggestions. In the revised manuscript, we have performed DEseq analysis using all ncRNAs. Then, we extracted significantly dysregulated ncRNAs using the cutoff criterion of P-value < 0.01 and $|\log_2FC| > 0.85$.

Subsequently, we identified two upregulated miRNAs, five downregulated miRNAs, zero upregulated piRNAs, and four downregulated piRNAs [mean read count > 100, area under the curve (AUC) > 0.8]. Finally, we found that miR-16-2-3p, miR-378a-3p, and miR-483-5p could be novel biomarkers.

4) Results of RNA-seq data

In Figure 1 (E_H), the heatmaps and PCA do not show two RNA profiles for pregnant and non-pregnant samples as the authors stated (Not for miRNAs and not for piRNAs). The authors seem to describe what they would like to see.

Given the PCA and heatmap plots that the authors showed, they could have discussed a potential grouping of samples in pregnant and non-pregnant with the exception of a few samples, which could be the ones shown in the Fig1 E (heatmap).

Suggestion: After performing another RNA-seq data analysis based on a more strict criterion and presenting only the DA RNAs in a PCA and heatmap, if there are still samples that they do not group together, they could be discussed as: 1) potential outliers or 2) maybe as samples that they could provide a favorable environment for an oocyte, which could have good quality but because other reasons (male, chance) these oocytes finally do not end in a successful pregnancy. Please, in PCA and heatmaps write the samples labels, which allow to identify which samples do not group together.

Answer: As mentioned above, we have completely re-performed the analysis. We performed heatmap and PCA analysis using differentially expressed ncRNAs, which were identified through volcano plots. The heatmap and PCA showed that the ncRNA profiles of the pregnant and non-pregnant groups were more clearly separated.

5) Selected miRNAs and piRNAs:

Did the authors consider the function of these selected miRNAs and piRNA and the potential target genes that could act in the oocyte and affects its quality and have an impact on the potential embryo? It is not discussed at all in the manuscript. For example, hsa-miR-483-5p, was previously found in human FF EVs (Martinez et al., 2018 10.1038/s41598-018-35379-3).

Answer: Thank you for providing this valuable reference. As suggested, we have incorporated it into the revised manuscript and carefully revised the Discussion to clarify the function of small ncRNAs related to oocyte quality (Page 14, Lines 344–361).

6) Results of cohort study (3 patients) and selection of biomarkers:

The heatmap in Fig.3 (A) does not show two different RNA profiles between pregnant and non-pregnant women (Even the clustering three does not group samples in 2 groups). Besides, the

results for the 3 selected RNAs showed a lot of variability among patients and do not reflect the results stated by the authors (Fig 3. (C)). It is difficult to see how they could be pointed as biomarkers, with these results and without discussing their potential functions and target genes. Again here, Why the authors do not illustrate only the DA miRNAs, which could show the biggest differences among samples.

Suggestion: If the goal is to suggest RNAs as biomarkers, another technique together with another cohort of samples, could have been used to validate the results of these selected miRNAs (for example by Taq-man assays for specific miRNAs)

Answer: The expression of these miRNAs (miR-16-2-3p, miR-30e-5p, miR-483-5p, miR-1246, and miR-1290) in FF-sEVs was confirmed by quantitative PCR. miR-16-2-3p and miR-30e-5p showed significant differences between the pregnant and non-pregnant groups, and the other three miRNAs showed no significant differences, but there were trends in expression levels (Supplementary Fig. S4A). Regarding piRNAs, there are no validated and commercially available assays; therefore, we only tested miRNAs for qPCR.

7) Data mining

This part is very superficial with only 5 pathways shown and with a very brief discussion. Suggestion: Other available online tools could be used to find potential target genes for the differentially abundant miRNAs and associated functional enrichment analysis. Combination of results from different functional enrichment analysis can confirm results or bring different potential perspectives.

Answer: Thank you for your helpful comment. We further analyzed the functions of the miRNAs revealed in our study and verified their relevance to embryo quality and follicular development. We have carefully revised the Results and Discussion as follows.

• Page 13, Lines 317–326 (Result section)

First, the miRSystem algorithm was used to assess the potential functions of the seven differentially expressed miRNAs (miR-16-2-3p, miR-204-3p, miR-378a-3p, miR-424-5p, miR-483-5p, miR-1246, and miR-1290) shown in Fig. 2E. Among these miRNAs, we focused on three selected miRNAs (miR-16-2-3p, miR-378a-3p, and miR-483-5p), which were found to be associated with 189 mRNAs according to miRSystem analysis. Second, we selected 640 mRNAs associated with embryo quality form a previous study (PMID: 36857405) and integrated 189 mRNAs related to the selected miRNAs, resulting in narrowed-down set of seven genes (Fig. 3D). These seven mRNAs included genes that play critical roles in embryonic development, such as SOX and RHO. Gene ontology (GO) terms were significantly dysregulated by the seven mRNAs, with notable impact on the BMP signaling pathway (Fig. 3D).

· Page 14, Lines 344–361 (Discussion section)

8) Comparison with other studies.

Did the authors compare the miRNAs and piRNAs identified in this study with others?

Which was the overlap? It is stated that variability among studies has been discussed in the manuscript (Lines 312 to 317), but it is not mentioned which comparison has been done and which small noncoding RNAs were found in common to other studies.

Answer: Thank you for your comment. We have carefully revised the Discussion to compare the previously published studies (Page 14, Lines 344–361).

9) Datasets uploaded to repository data Datasets should be uploaded to repository data (for example: <https://www.ncbi.nlm.nih.gov/geo/>).

Answer: We have uploaded our datasets as accession No. GSE 221271.

10) Methodology:

Methodology should be described in a way that experiments or techniques can be repeated by others. For example, description of preparation of samples for CryoTEM analysis should be improved or references added.

Answer: We apologize for the insufficient descriptions. We have carefully revised the Methods section of Cryo-TEM (Page 8, Lines 169–184).

11) Figures legend: Legend should be written with all necessary information to understand the heatmap and PCA. For example: What is z-score?

Answer: In the context of heatmap, the z-score is a statistical measure used to standardize and compare values within a dataset, enabling the identification of relatively high or low values and patterns in the data. Because this notation is very common, we avoid providing details in the legends.

Minor concerns:

-Some sentence needs to be re-written: “EVs are known to be the cargo of several mRNAs, proteins, and miRNAs”. Please change to “are known to contain...”

Answer: We apologize for this typographical error. We have now corrected it.

-Please change small RNA for small non coding RNAs (small ncRNAs) across the manuscript.

Answer: Thank you for your insightful comment. A non-coding RNA (ncRNA) is a functional RNA

molecule that is not translated into a protein. In our small RNA sequencing, we selectively isolated small RNAs (<100 base pairs) by cutting the gel, but we did not apply the specific procedure to enrich ncRNAs. In this study, we mapped raw read sequences to miRNA and piRNA references. ncRNA are the major class of small RNAs, but they are not perfectly equal (PMID: 18270516). In addition, fragmented coring RNAs might contaminate the samples, but this concern was successfully eliminated during the mapping process. We used the term small RNA sequencing in this manuscript, despite fully respecting the comment.

Reviewers' comments:

Reviewer #1 (Remarks to the Author):

In the presented study, the authors propose using follicular fluid-derived small EVs (and thereby their content) as biomarkers to predict the chances of success for patients undergoing fertility treatment in assisted reproductive technology. The current method of measuring the Anti-Müllerian hormone, which indicates the ovarian reserve, is an insufficient and possibly incorrect parameter to predict the outcome of the fertility treatment. Hence, the authors claim that there is a need for better markers to help clinicians and patients realistically assess the chances of success in the event of repeated ART treatments, multiple times.

The authors have tried to address the previous comments and revise the manuscript, however there are still some major issues. Here are a few points which I think need to be considered before publication:

148: if mentioning rpm, it is useful to the reader to know the rotor radius as well. Otherwise report the speed in g(relative centrifugal force). The way it has been reported on line 156 is correct.

199: The authors should mention which protein ladder was used so it's easier for the reader to compare the sample bands against it. Also, the wb images in the supplementary figure 1 have no labelling for each well so it is hard to assess what is what. Ideally, one well should be the ladder and then the samples. Also, the wb should have been done to compare cell lysate vs EV sample (as it is conventionally done in a lot of EV related publications). The authors have shown it for the GRP sample but I am curious why it was not included in the other blots. This would help the reader appreciate the enrichment of the specified markers on the EV sample OR the lack of GRP signals in the EV samples compared to the cell lysates. It would also be useful to mention what dilutions of the 1^oantibody were used for the WB. (e.g. – 1:100, 1:1000, 1:10,000 etc).

312: The authors make a statement that – ‘ low levels of miR-378 and miR-483 were possible biomarkers for pregnancy. In the next line high levels of miR 166-2-3P are associated with pregnancy ? It is confusing. I think what they mean is that – the decreased expression of 378 and 483 , meanwhile the increased expression of 16-2-3p can be linked to ‘pregnancy’ outcome or can serve as a biomarker ?

313: The quality of IVF embryos is directly related to the pregnancy outcomes

Overall, the study lacks functional data as such, which I think is necessary to drive the point home that these FF-EV's are suitable sources of biomarkers especially with regards to the outcome in fertility treatments. In figure 3, where they compare the 3 patients which had pregnant as well as non pregnant oocytes, how can one assess if the lowered gene expression is the cause for the improved chances of pregnancy or are the observed differences the result of pregnancy occurring due to some other factor? The authors have also not discussed or proposed a way in which this knowledge could be utilized to improve the outcome of ART. Would this process of obtaining such FF samples be viable? How could it be utilized by medical practitioners for example?

The authors themselves admit that some of their results are the exact opposite of what others in the field have found. Unfortunately, this important difference has been attributed to factors such as technical differences or methodologies used. I believe in this case the study should have been

subjected to much more rigor.

Reviewer #2 (Remarks to the Author):

The manuscript “Follicular fluid-derived small extracellular vesicles for predicting successful fertility treatment in assisted reproductive technology” investigated a possible association between follicular fluid miRNAs and pregnancies outcomes. The manuscript is well written and points to three miRNAs as good predictors of pregnancy outcomes. I have a couple of concerns regarding the sample description and follicular fluid origin, that can change the interpretation of the manuscript.

Concerns:

- Is not clear to me how authors treated individuals with multiple follicles. Each follicle become a sample? In the spectrum of 10 FF samples, how many individuals were used? Did you have follicles within the same individual that had different outcomes (pregnant and non-pregnant)?
- Please verify the text in lines 260-263. Samples are from different follicles within the same individuals or different follicles from different individuals?
- Figure 1: According to the western blot images, all the EVs preparations have a some cell contamination. Since authors did not clean the samples before freezing, this can be a cell death contamination due to freezing the samples. It is recommended to perform the differential centrifugation before freezing the samples, to avoid cell damage during the freezing process. The TEM picture only show individual vesicles, could you add a lower magnification image demonstrating different vesicles within the same sample?
- Table 1 is confusing. I am having a hard time to understand the number of samples, number of patients, and number of follicles used from each patient. Please clarify this within your material and methods.
- Can you please explain the biological connection between a EV-miRNA and an embryo quality associated gene? This type of association does not make sense if we are thinking about the role of miRNAs within cells and in the EVs.
- Please verify the sentence within lines 326-328. Can you explain this possible association and how this can be biologically possible?

Q1) Assessment of RNA quality: Were the EV RNA profiles analyzed by Bioanalyzer to assess the similarity of RNA profiles among pregnant and non-pregnant groups and in general across 7 samples? (It seems only RNA concentration by Qubit was analyzed).

Answer: We apologize for the inadequate data. RNA concentrations were measured using a Bioanalyzer and all data were added to Supplementary Fig. S4B. While RNA concentrations were low, these results are reasonable because they detected small non-coding RNAs in the FF-sEVs. In addition, only a limited amount of FF sample could be collected, however, RNA sequencing has the advantage that RNA concentration can be amplified for analysis. Even RNA in FF at low concentrations can be a target for study by sequencing using amplification methods.

Response: Assessment of RNA quality had to be done with Bioanalyzer to check the RNA quality with RIN (RNA Integrity Number) to see if it was degraded or not. This would be critical to know since the RNA is to be further used for the library prep. The supplementary figure 4B does not show that.

Q2) Which was the total RNA amount was used as starting material for the library preparation?

Answer: We apologize for the inadequate data. As shown in the additional data on bioanalyzers, RNA concentrations were approximately 0.1–1 ng/μL, and we used 6 μL of RNA for library preparations. We understand that this concentration is low, but we confirmed that sequencing worked well for analysis based on replication assessments (data not shown).

Response: The authors mention that 6ul of the samples were used for the library prep but how did they normalize across the samples? 6ul of sample 1 may give 6ug but from sample 2 may give 18ug? So, the authors do not provide a satisfactory answer to the question.

Q3) All miRNAs and piRNAs detectable or identified in the study should be provided in the manuscript. Besides, every list showing the filtering of the miRNA and piRNA data together with the final lists of RNA showing higher or lower abundance after statistical analysis (FDR, FC, P-values) should be provided. The manuscript only shows 2 lists of selected miRNAs and piRNAs (with a very unclear filtering criteria).

Answer: Thank you for your comment. As shown in Supplementary Fig. S2, we annotated 1,212 miRNAs and 1,076,849 piRNAs. We can provide both miRNA and piRNA profiles. However, the piRNA profile was less informative because the read counts of most piRNAs were very low. Instead, we have provided a list of selected miRNAs and piRNAs as Supplementary Tables, including the average read count of the pregnancy and non-pregnancy groups, area under the ROC curve, and P-value calculated using DEseq2.

Response: Comment not addressed adequately. The miRNA profiles can be provided. ..What were the reads aligned to? This has not been mentioned in the methods.

Q4) Are the authors aware of studies done by others or have they considered using samples from normal fertile females as a control against all these females who are undergoing treatment? ART involves hormone regulation using drugs and would that have an influence on the cargo of these sEVs? Reference 39 addresses this issue with regards to the expression of miR-320a-3p..

Q5) Did the authors consider the function of these selected miRNAs and piRNA and the potential target genes that could act in the oocyte and affects its quality and have an impact on the potential embryo? It is not discussed at all in the manuscript. For example, hsa-miR-483-5p, was previously found in human FF EVs (Martinez et al., 2018 10.1038/s41598-018-35379-3).

Answer: Thank you for providing this valuable reference. As suggested, we have incorporated it into the revised manuscript and carefully revised the Discussion to clarify the function of small ncRNAs related to oocyte quality (Page 14, Lines 344–361).

Response: The key point here is that miR-483-5p was found to be expressed in FF EV samples of oocytes which fertilized vs ones which did not. In the present study it seems to be downregulated. Do the authors have any comments on that?

Q6) P14, Line 341: association between pregnancy outcomes and FF-EV-derived small ncRNAs has not been reported.

Response: The results are different, but the aim of this study overlaps with the one published by Martinez et al.

Q7) p12, Line 296-297 The authors say that based on the qPCR results the expression of miR-16-2-3p and miR-30e-5p was significantly higher.. however, the authors used students t-test to calculate the significance of the p-value when clearly the data distribution indicates the need for a non-parametric test.

Conclusion: The authors have addressed most of the previous comments however the manuscript in the current format fails to convince why this study is relevant or important given the body of similar research work done by others. The authors could have accessed published datasets to compare and contrast their findings.

Reviewers' comments:

Reviewer #1 (Remarks to the Author):

In the presented study, the authors propose using follicular fluid-derived small EVs (and thereby their content) as biomarkers to predict the chances of success for patients undergoing fertility treatment in assisted reproductive technology. The current method of measuring the Anti-Müllerian hormone, which indicates the ovarian reserve, is an insufficient and possibly incorrect parameter to predict the outcome of the fertility treatment. Hence, the authors claim that there is a need for better markers to help clinicians and patients realistically assess the chances of success in the event of repeated ART treatments, multiple times.

The authors have tried to address the previous comments and revise the manuscript, however there are still some major issues. Here are a few points which I think need to be considered before publication:

Thank you for your positive feedback on our data. Based on your suggestions, we have thoroughly revised the article. We focused on extracellular vesicles present in follicular fluid (FF-EVs). One of the strengths of our study is setting the endpoint as successful pregnancies, whereas other studies have solely assessed embryo quality. Furthermore, we have included the functional analysis of miRNAs revealed in our study and verified their relevance to embryo quality and follicular development. Additionally, we have mentioned the clinical usage of our findings. These findings are worth to share and move this fields forward.

148: if mentioning rpm, it is useful to the reader to know the rotor radius as well. Otherwise report the speed in g(relative centrifugal force). The way it has been reported on line 156 is correct.

Answer: Thank you for your comment. We have carefully revised the unit rpm to $\times g$ (Page 7, Line 148 and Line 154).

199: The authors should mention which protein ladder was used so it's easier for the reader to compare the sample bands against it. Also, the wb images in the supplementary figure 1 have no labelling for each well so it is hard to assess what is what. Ideally, one well should be the ladder and then the samples. Also, the wb should have been done to compare cell lysate vs EV sample (as it is conventionally done in a lot of EV related publications). The authors have shown it for the GRP sample but I am curious why it was not included in the other blots. This would

help the reader appreciate the enrichment of the specified markers on the EV sample OR the lack of GRP signals in the EV samples compared to the cell lysates. It would also be useful to mention what dilutions of the 1st antibody were used for the WB. (e.g. – 1:100, 1:1000, 1:10,000 etc).

Answer: We sincerely apologize for the confusion. We used the MagicMark™ XP Western Protein Standard (Thermo Fisher Scientific) as a protein ladder and revised the Methods section (Page 9, Lines 198). And we added the dilutions of the 1st antibody (CD63; 1/1000, CD9, CD81 and GRP94; 1/100) in the Methods section (Page8, Lines 191–194). We carefully revised Supplementary Figure S2 (in the revised version) to add the labeling of each well.

In general, control of cell lysates can be set at cell line experiment. In this study, all experiments were done using follicular fluids, which do not have companion cells. As you can see in Supplementary Figure S2, cell lysate of fibroblasts was used as the positive control.

312: The authors make a statement that – ‘ low levels of miR-378 and miR-483 were possible biomarkers for pregnancy. In the next line high levels of miR 166-2-3P are associated with pregnancy ? It is confusing. I think what they mean is that – the decreased expression of 378 and 483 , meanwhile the increased expression of 16-2-3p can be linked to ‘pregnancy’ outcome or can serve as a biomarker ?

Answer: We sincerely apologize for the confusion. In the Figure 3C, we compared the miRNA expression in FF-sEV of each pregnant and non-pregnant oocytes in the same individual. All three patients showed lower expression levels of miR-378a-3p and miR-483-5p in pregnant FF samples compared to non-pregnant FF samples. On the other hand, miR-16-2-3p showed higher expression levels in pregnant FF samples than non-pregnant FF samples in Patient 2 and Patient 3. These miRNA expression patterns may capture the characteristics of oocytes that conceive beyond the individual patient. We added the arrows to Figure 3C for clarity.

313: The quality of IVF embryos is directly related to the pregnancy outcomes

Overall, the study lacks functional data as such, which I think is necessary to drive the point home that these FF-EV's are suitable sources of biomarkers especially with regards to the outcome in fertility treatments. In figure 3, where they compare the 3 patients which had pregnant as well as non pregnant oocytes, how can one assess if the lowered gene expression is the cause for the improved chances of pregnancy or are the observed differences the result of pregnancy occurring due to some other factor? The authors have also not discussed or proposed a way in which this knowledge could be utilized to improve the outcome of ART. Would this process of obtaining such FF samples be viable? How could it be utilized by medical practitioners for example?

The authors themselves admit that some of their results are the exact opposite of what others in the field have found. Unfortunately, this important difference has been attributed to factors such as technical differences or methodologies used. I believe in this case the study should have been subjected to much more rigor.

Answer: Although the existing embryo grading system could not predict the pregnancy outcome, we believe it is valid and differentiates the present study from other studies in that the outcome is pregnancy, taking maternal factors into account. In this study, we found the clinically available biomarkers in FF which could be sampled at the time of oocyte collection. In the ART laboratory, we could obtain miRNA expression of FF-sEVs by qPCR rapidly. These biomarkers could predict pregnancy outcome and may assist in deciding which embryos to transfer when multiple fertilized oocytes are obtained preferentially. We understand such differences can happen, and all studies should be rigorously performed. It is not always best that the data be the same as previous works. All methods were sincerely disclosed in the manuscript, and we hope our study can contribute to improving the understanding of this field.

Reviewer #2 (Remarks to the Author):

The manuscript “Follicular fluid-derived small extracellular vesicles for predicting successful fertility treatment in assisted reproductive technology” investigated a possible association between follicular fluid miRNAs and pregnancy outcomes. The manuscript is well written and points to three miRNAs as good predictors of pregnancy outcomes. I have a couple of concerns regarding the sample description and follicular fluid origin, that can change the interpretation of the manuscript.

Thank you for your valuable feedback. In response to your comments, we added detailed explanations and have carefully revised the article.

Concerns:

- Is not clear to me how authors treated individuals with multiple follicles. Each follicle become a sample? In the spectrum of 10 FF samples, how many individuals were used?

Did you have follicles within the same individual that had different outcomes (pregnant and non-pregnant)?

Answer: We sincerely apologize for the confusion. As you mentioned, each follicle becomes a sample because we could collect each FF individually. We added Supplementary Figure S1 (in the revised version), which indicates that patient No., FF sample No., and pregnant or non-pregnant cases.

- Please verify the text in lines 260-263. Samples are from different follicles within the same individuals or different follicles from different individuals?

Answer: As mentioned above, FF samples were taken from different follicles within the same individuals for Patient No. 1 through No. 3 and from different follicles in different individuals for Patient No.4 through No. 19.

- Figure 1: According to the western blot images, all the EVs preparations have a some cell contamination. Since authors did not clean the samples before freezing, this can be a cell death contamination due to freezing the samples. It is recommended to perform the differential centrifugation before freezing the samples, to avoid cell damage during the freezing process. The TEM picture only show individual vesicles, could you add a lower magnification image demonstrating different vesicles within the same sample?

Answer: We routinely store follicular fluids after removing cells by 430 x g centrifugation. As

described in the method sections, we performed differential and serial centrifugation methods. Therefore, the potential problems of the freezing process should be resolved. However, the purity of recovered EVs is always essential, and it will be carefully considered. Ideally, we wanted to see multiple vesicles in TEM, but the concentration of FF-EVs need to be higher to observe in TEM. We have added the low magnification images in Fig 1D, but different vesicles were not visualized. In additional TEM images in Fig 1E, the conditions were the same to cryo-TEM in Fig 1D. We aimed to check the structure of EVs by TEM, and the distribution of vesicles can be checked in Fig 1B.

- Table 1 is confusing. I am having a hard time to understand the number of samples, number of patients, and number of follicles used from each patient. Please clarify this within your material and methods.

Answer: We sincerely apologize for the confusion. We added Supplementary Figure S1, which indicates that patient No. (n = 19), FF sample No. (n = 24), and pregnant or non-pregnant cases (pregnant cases; n =10, non-pregnant cases; n=14). We also revised the Methods section, including the sample correlation (Page 6, Line 120).

- Can you please explain the biological connection between a EV-miRNA and an embryo quality associated gene? This type of association does not make sense if we are thinking about the role of miRNAs within cells and in the EVs.

Answer: As described in the revised manuscript, we referred to previous works on which types of genes can affect embryo quality. We hypothesized that the molecular profiles can be affected by the condition of granulosa cells, and FF-EV can contain granulosa cell-derived EVs. The work focused on the gene expression of granulosa cells from the good-quality embryo group to the poor-quality embryo group. Six hundred forty differentially expressed genes were detected, and related to the biological processes, molecular functions, and cellular components. Among them, seven genes were picked up as embryo quality-related genes by merging our selected 189 genes which are associated with the target miRNAs in our study. EV-miRNAs are known to have specific functional communications with surrounding cells. EV-miRNA influences the expression level of the target gene and might associate with oocyte development. By the way, this study is aimed to discover the biomarkers to predict pregnancy, and we will assess further detailed functions in the future.

- Please verify the sentence within lines 326-328. Can you explain this possible association and how this can be biologically possible?

Answer: These seven genes were associated with selected miRNAs and involved in embryo

quality. In particular, the BMP signaling pathway, which is essential for oocyte development, was extracted in the GO analysis. These miRNAs may contribute to oocyte development by regulating gene expression in embryo quality. According to your comment, we revised the sentence in lines 327-329.

Reviewer #3 (Reviewer #1 were responded instead of Reviewer #3):

Q1) Assessment of RNA quality: Were the EV RNA profiles analyzed by Bioanalyzer to assess the similarity of RNA profiles among pregnant and non-pregnant groups and in general across 7samples? (It seems only RNA concentration by Qubit was analyzed).

Answer: We apologize for the inadequate data. RNA concentrations were measured using a Bioanalyzer and all data were added to Supplementary Fig. S4B. While RNA concentrations were low, these results are reasonable because they detected small non-coding RNAs in the FF-sEVs. In addition, only a limited amount of FF sample could be collected, however, RNA sequencing has the advantage that RNA concentration can be amplified for analysis. Even RNA in FF at low concentrations can be a target for study by sequencing using amplification methods.

Response: Assessment of RNA quality had to be done with Bioanalyzer to check the RNA quality with RIN (RNA Integrity Number) to see if it was degraded or not. This would be critical to know since the RNA is to be further used for the library prep. The supplementary figure 4B does not show that.

Answer: As you pointed out, the RIN value is vital for understanding RNA quality. However, most EV researchers know the RNA size in small EVs is always short, which means the RIN value is low. We have tested, and in our samples, the data was the same. This is one of the reasons why we focused on small RNAs in EVs. However, it is possible to amplify and analyze a little amounts of RNA using next generation sequencing technology. We have added the following statement " RIN values of FF-EV RNAs were around 1.0-3.0" on Page 9, Line 209.

Q2) Which was the total RNA amount was used as starting material for the library preparation?

Answer: We apologize for the inadequate data. As shown in the additional data on bioanalyzers, RNA concentrations were approximately 0.1–1 ng/μL, and we used 6 μL of RNA for library preparations. We understand that this concentration is low, but we confirmed that sequencing worked well for analysis based on replication assessments (data not shown).

Response: The authors mention that 6ul of the samples were used for the library prep but how did they normalize across the samples? 6ul of sample 1 may give 6ug but from sample 2 may give 18ug? So, the authors do not provide a satisfactory answer to the question.

Answer: We have an assortment of RNA amounts with starting volumes. During the library preparation step, cDNA concentration was equalized among the samples.

Q3) All miRNAs and piRNAs detectable or identified in the study should be provided in the manuscript. Besides, every list showing the filtering of the miRNA and piRNA data together with the final lists of RNA showing higher or lower abundance after statistical analysis (FDR, FC, P-values) should be provided. The manuscript only shows 2 lists of selected miRNAs and piRNAs (with a very unclear filtering criteria).

Answer: Thank you for your comment. As shown in Supplementary Fig. S2, we annotated 1,212 miRNAs and 1,076,849 piRNAs. We can provide both miRNA and piRNA profiles. However, the piRNA profile was less informative because the read counts of most piRNAs were very low. Instead, we have provided a list of selected miRNAs and piRNAs as Supplementary Tables, including the average read count of the pregnancy and non-pregnancy groups, area under the ROC curve, and P-value calculated using DEseq2.

Response: Comment not addressed adequately. The miRNA profiles can be provided. What were the reads aligned to? This has not been mentioned in the methods.

Answer: We apologize for the confusing answer. In the revision, we provided all profiles of selected miRNAs and piRNAs in Supplementary Tables. In addition, raw data has been uploaded to the GEO database. Therefore, all readers can refer to all small RNA profiles. We described alignment as “the data were mapped to miRBase 22 and piRBase v3.0, allowing up to two mismatches” in the method section, which should be enough.

Q4) Are the authors aware of studies done by others or have they considered using samples from normal fertile females as a control against all these females who are undergoing treatment? ART involves hormone regulation using drugs and would that have an influence on the cargo of these sEVs? Reference 39 addresses this issue with regards to the expression of miR-320a-3p.

Answer: ART is an invasive procedure performed exclusively on infertile patients; FF can only be obtained from infertile patients who undergo oocyte retrieval on ART. This means it is ethically impossible to obtain FF from a woman with normal fertility because it is an invasive procedure. As you mentioned, hormonal treatment may influence the contents of EVs. However, since all of our study subjects are undergoing hormone therapy, we believe that is unlikely to be a problem.

Q5) Did the authors consider the function of these selected miRNAs and piRNA and the potential target genes that could act in the oocyte and affects its quality and have an impact on the potential embryo? It is not discussed at all in the manuscript. For example, hsa-miR-483-5p, was previously found in human FF EVs (Martinez et al., 2018 10.1038/s41598-018-35379-3).

Answer: Thank you for providing this valuable reference. As suggested, we have incorporated it into the revised manuscript and carefully revised the Discussion to clarify the function of small

ncRNAs related to oocyte quality (Page 14, Lines 344–361).

Response: The key point here is that miR-483-5p was found to be expressed in FF EV samples of oocytes which fertilized vs ones which did not. In the present study it seems to be downregulated. Do the authors have any comments on that?

Answer: The most important point is that this study was designed as an unbiased exploratory investigation. We already mentioned that the expression level of miR-483-5p is different than the previous report. However, it is difficult to determine which should be true because the examination methods and patient background influenced the results. We disclosed all detailed methods and hope our data can also be a key report on this field.

Q6) *P14, Line 341: association between pregnancy outcomes and FF-EV-derived small ncRNAs has not been reported.*

Response: The results are different, but the aim of this study overlaps with the one published by Martinez et al.

Answer: We have carefully rechecked the reference, but they aimed to compare the quality of oocytes. The overlapped part was too small to say this work is similar to the report. Currently, our study applied a unique approach, providing direct evidence that miRNA-based biomarkers can be helpful in the clinic. In this manuscript, we have already included the reference (Martinez's report) as #25 in the discussion section.

Q7) *p12, Line 296-297* The authors say that based on the qPCR results the expression of miR-16-2-3p and miR-30e-5p was significantly higher.. however, the authors used students t-test to calculate the significance of the p-value when clearly the data distribution indicates the need for a non-parametric test.

Answer: Our statistician claimed that the sample of this study is not satisfied to determine the distribution of samples. The sample size is limited, and difficult to conclude whether parametric or not. Regarding this data, this is not necessarily wrong.

Conclusion: The authors have addressed most of the previous comments however the manuscript in the current format fails to convince why this study is relevant or important given the body of similar research work done by others. The authors could have accessed published datasets to compare and contrast their findings.

Answer: We focused on extracellular vesicles present in follicular fluid (FF-EVs). One of the unique and significant points in our study is the endpoint, which is set as successful pregnancy. In contrast, other studies have solely assessed embryo quality, which does not always predict a successful outcome. Additionally, we focused on the small RNAs, including piRNAs, which have

been reported to work in germline cells. As mentioned above, our primary outcome is pregnancy, so accessing published datasets and analyzing the results of other studies is difficult. At the same time, we totally respect previous works and included discussion mentioning such works as much as we can.

Reviewers' comments:

Reviewer #1 (Remarks to the Author):

In the presented study, the authors propose using follicular fluid-derived small EVs (and thereby their content) as biomarkers to predict the chances of success for patients undergoing fertility treatment in assisted reproductive technology. The current method of measuring the Anti-Müllerian hormone, which indicates the ovarian reserve, is an insufficient and possibly incorrect parameter to predict the outcome of the fertility treatment. Hence, the authors claim that there is a need for better markers to help clinicians and patients realistically assess the chances of success in the event of repeated ART treatments, multiple times.

The authors have tried to address the previous comments from the reviewers and revised the manuscript yielding a much better and understandable version. I do feel like some of their rebuttal comments were better explained than the actual text in the manuscript and it would help them to add those explanations into the main text.

Line 231: It would help if they can provide a slightly in-depth explanation of why they have 299 oocyte retrievals when the actual number of patients in the study is 47; 33P (minus the excluded 23 samples) + 14 NP. My understanding is that each patient probably has 5 or more oocytes. To someone not familiar with how ART works it would help.

Line 397: Inherently the procedure to obtain these FF-EVs is also invasive (which is why the authors argued that normal / healthy females were excluded since they would require permits). So, in the discussion when they mention that use of FF-EVs can contribute to non-invasive pregnancy predictors it causes confusion. My understanding is that since the oocyte retrieval already grants access to the FF-EV sample, it is a useful by-product of the procedure itself which works as a predictor.

Reviewer #2 (Remarks to the Author):

Dear authors,
Thank you for answering all of my concerns.

Reviewer #3 (Remarks to the Author):

Comments to the Author

The authors analyzed non-coding RNAs (ncRNAs) in follicular fluid (FF) to search for factors that predict pregnancy rate (and/or live birth rate). After successfully isolating extracellular vesicles, the authors performed small RNA sequencing with extracted RNA to examine the expression of ncRNAs (miRNAs and piRNAs). They detected miR-16-2-3p, miR-378a-3p, and miR-483-5 p as a valid predictor for the establishment of pregnancy.

Since there is a large gap between the timing of ncRNA expression and the timing of the establishment of pregnancy and live birth, it is not known whether there is a direct causal

relationship (although the current study is an observational study). Therefore, it would be desirable to have a more robust functional assay (e.g., loss-of-function assay using animal or cellular models). Below are my comments that might help the authors improve the current study.

Major point

1. It seems normal to assume that ncRNAs in follicular fluid are most effective for oocyte maturation and fertilization. ncRNA expression differences may correlate with multiple factors other than pregnancy, which is the subject of this study, including follicle size, oocyte maturation rate, fertilization rate, embryo development rate, etc. As a logical jump in this paper, it should be pointed out that there is a large timing gap between when miRNAs are expressed and when they reach pregnancy and delivery. Therefore, the correlation between ncRNAs expression and the establishment of pregnancy could be a chance correlation.

2. Figure 3D

Which cells are the miRNAs identified in this analysis derived from? It has been reported that BMP-2 and -6 are expressed in granulosa cells, BMP-4 and -7 in capsular and stromal cells, and BMP-6, -15, and GDF-9 in oocytes.

LINES 318-331

3. miRNAs usually bind to target mRNAs and are involved in post-transcriptional regulation of gene expression. It is believed that miRNAs bind mainly to the 3'UTR of the target mRNA, degrading the mRNA and repressing translation. It has also been reported that mRNA binds to 5'UTRs and promoter regions other than 3'UTRs, and promotes or inhibits transcription in addition to translation inhibition. Based on the above, it would be great to analyze the expression of mRNAs in the follicular fluid to see what kind of differences there are in mRNA transcription.

4. In animal oocytes, retrotransposons that reactivate in oocytes cause oocyte-specific and species-specific DNA methylation. With this in mind, can you verify how the piRNAs identified in this study regulate transposon expression, along with analysis of expression levels?

Minor point

5. Which contributes more to the difference in ncRNAs in follicular fluid, the presence or absence of an established pregnancy, or the difference in population? Since humans are a heterogeneous population, such a comparison can be useful.

6. Is there a correlation between follicle size oocyte maturation rate and ncRNA expression levels?

7. Fig.1C

It would be preferable to show the entire Membrane (is Fig. S2 responsible for that?).

8. Fig. 3A

The P5 data is confusing because it appears to be derived from both Patient 1 and Patient 2.

9. LINES 110-121

Please provide definitions of "pregnant" and "non-pregnant". Consider the possibility that different miRNAs/piRNAs may be extracted from chemical pregnancies, clinical pregnancies, and childbirth.

10. LINES 137-142 Method section: Embryo transfer

Please state the grade in the Gardner classification of the blastocysts used for embryo transfer.

11. LINES 144-149 Method section: Sample collection

If many follicles were aspirated at once, cross-contamination between follicular fluids could make evaluating ncRNAs contained in single follicles difficult. Please elaborate in the method section on how you avoided cross-contamination (change the needle each time at an oocyte aspiration, wash, etc.).

12. Fig.1A, LINES 363-374 Discussion

It is debatable whether this ncRNA-based analysis goes beyond or complements the classical Gardner classification since by design it compares embryos that have developed above a “certain” grade (Fig. 1A). As mentioned in the discussion, the authors are acting modestly as a complementary position. On the other hand, in the clinical setting, there are fertilized eggs that are euploid and result in delivery, even if they are of low grade based on Gardner criteria. The analysis of “low-grade” samples may be expected to go beyond conventional evaluation methods.